# Density of states prediction for materials discovery via contrastive learning from probabilistic embeddings

Shufeng Kong [1,6], Francesco Ricci[2,6], Dan Guevarra[3], Jeffrey B. Neaton [2,4,5 ✉], Carla P. Gomes [1 ✉] & John M. Gregoire [3 ✉]

Machine learning for materials discovery has largely focused on predicting an individual scalar rather than multiple related properties, where spectral properties are an important example. Fundamental spectral properties include the phonon density of states (phDOS) and the electronic density of states (eDOS), which individually or collectively are the origins of a breadth of materials observables and functions. Building upon the success of graph attention networks for encoding crystalline materials, we introduce a probabilistic embedding generator specifically tailored to the prediction of spectral properties. Coupled with supervised contrastive learning, our materials-to-spectrum (Mat2Spec) model outperforms state-of-the-art methods for predicting ab initio phDOS and eDOS for crystalline materials. We demonstrate Mat2Spec's ability to identify eDOS gaps below the Fermi energy, validating predictions with ab initio calculations and thereby discovering candidate thermoelectrics and transparent conductors. Mat2Spec is an exemplar framework for predicting spectral properties of materials via strategically incorporated machine learning techniques.

[1] Department of Computer Science, Cornell University, Ithaca, NY, USA. [2] Material Science Division, Lawrence Berkeley National Laboratory, Berkeley, CA, USA. [3] Division of Engineering and Applied Science, California Institute of Technology, Pasadena, CA, USA. [4] Department of Physics, University of California, Berkeley, Berkeley, CA, USA. [5] Kavli Energy NanoSciences Institute at Berkeley, Berkeley, CA, USA. [6] These authors contributed equally: Shufeng Kong, Francesco Ricci. ✉email: jbneaton@lbl.gov; gomes@cs.cornell.edu; gregoire@caltech.edu

Spectral properties are ubiquitous in materials science, characterizing properties ranging from crystal structure (e.g., X-ray absorption and Raman spectroscopy), the interactions of material with external stimuli (e.g., dielectric function and spectral absorption), and fundamental characteristics of its quasi-particles (e.g., phonon and electronic densities of states)[1]. Matching the breadth of spectral properties is the breadth of methods to characterize them, traditionally experimental and ab initio computational techniques, which are central to materials discovery and fundamental research. Materials discovery efforts are largely defined by searching for materials that exhibit specific properties, and acceleration of materials discovery has been realized with automation of both experiments and computational workflows[2,3]. These high throughput techniques have been effectively deployed in a tiered screening strategy wherein high-speed methods that may sacrifice some accuracy and/or precision can effectively down-select materials that merit detailed attention from more resource-intensive methods[4–6]. The recent advent of machine learning prediction of materials properties has introduced the possibility of even higher throughput primary screening due to the minuscule expense of making a prediction for a candidate material using an already-trained model[7–10]. Toward this vision, we introduce the materials to spectrum (Mat2Spec) framework for predicting spectral properties of crystalline materials, demonstrated herein for the prediction of the ab initio phonon and electronic densities of state.

Successful development of machine learning models for materials property prediction hinges upon encoding of structure-property relationships to provide predictive models that serve as primary screening tools and/or provide scientific insights via inspection of the trained model[9,11,12]. Accelerated screening of candidate materials is especially important when a dilute fraction of materials exhibits the desired properties, as is inherently the case when searching for exemplary materials. Primary screening constitutes a great opportunity for machine learning (ML)-accelerated materials discovery but is challenged by the need for models to generalize, both in predicting the properties of never-before-seen materials and in predicting property values for which there are few training examples[13,14].

Most of the ML models developed in computational materials science to date have been focused on individual scalar quantities. Common properties are those directly computed ab initio, such as the formation energy[11,15,16], the shear- and bulk-moduli[11,15,17,18], the band gap energy[11,16,18,19], and the Fermi energy[11]. Additional targets include properties calculated from the ab initio output, which are typically performance metrics for a target application such as Seebeck coefficient[20,21]. For a complete review of target properties predicted via ML, see ref. [22]. Efforts to featurize materials for use in any prediction task started with an engineered featurizer algorithm[23] and evolved to automatic generation of features via aggregation from multiple sources[24]. This evolution mirrors the best practices in machine learning that have evolved from feature engineering to internalizing feature representation in a model trained for a specific prediction task. This is reflected in materials property prediction with models that generate a latent representation of a material from its composition and structure, for which graph neural networks (GNN) are the state of the art due to their high representation learning capabilities[25].

CGCNN[26] is among the first graph neural networks proposed for materials property prediction. CGCNN encodes the crystal structures as graphs where the unit cell of the crystal material is represented as a graph such that nodes would represent the atoms and connecting edges would represent the bonds shared among the atoms. MEGNet[12] expanded this concept by introducing a global state input including temperature, pressure, and entropy. The state of the art is well represented by GATGNN[9], which uses local attention layers to capture properties of local atomic environments and then a global attention layer for weighted aggregation of all these atom environment vectors. The expressiveness of this model allows it to outperform previous models in single-property prediction. Lastly, another different approach is demonstrated by AMDNet[16] which uses structure motifs and their connections as input of a GNN to predict electronic properties in metal oxides.

While multi-property prediction has been addressed only recently, rapid progress has been made by building upon the foundation of the single-target prediction models. De Breuck et al.[27] presented MODNet and highlighted the benefit of using feature selection and joint learning in materials multi-property prediction with a small dataset.

Broderick and Rajan[28] used principle component analysis to generate low-dimensional representations of eDOS enabling its prediction for elemental metals. Yeo et al.[29] expanded this approach to metal alloys and their surfaces using engineered features to interpolate from simple metals to their alloys. Del Rio et al.[30] proposed a deep learning architecture to predict the eDOS for carbon allotropes. Mahmoud et al.[31] presented a ML framework based on sparse Gaussian process regression, a SOAP-based representation of local environment, and an additive decomposition of the electronic density of states to learn and predict the eDOS for silicon structures. Bang et al.[32] applied CGCNN to eDOS, compressed via principle component analysis, of metal nanoparticles.

These initial demonstrations of eDOS prediction focus on specific classes of materials with a limited structural and chemical diversity, making them ill-suited for the present task of generally-applicable prediction of phDOS and eDOS. The state of the art method that is most pertinent to the present work is the E3NN model that was recently demonstrated for predicting phDOS[14]. E3NNs are Euclidean neural networks, which by construction include 3D rotations, translations, and inversion and thereby capture full crystal symmetry, and achieve high-quality prediction using a small training set of $\sim 10^3$ examples. E3NN obtains very good performance in predicting the computed phDOS from only the crystal structure of materials. To the best of our knowledge, GNN-based methods have not been reported for phDOS or eDOS prediction of any crystalline material. To create a baseline GNN model in the present work, we adapt GATGNN[9] for spectrum prediction. We recently reported a multi-property prediction model H-CLMP[7] for the prediction of experimental optical absorption spectra from only materials composition. H-CLMP implements hierarchical correlation learning by coupling multivariate Gaussian representation learning in the encoder with graph attention in the decoder.

In the present work, we introduce Mat2Spec, which builds upon the concepts of H-CLMP to address the open challenge of spectral property prediction with a GNN materials encoder coupled to probabilistic embedding generation and contrastive learning. Mat2Spec is demonstrated herein for eDOS and phDOS spectra prediction for a broad set of materials with periodic crystal structures from the Materials Project[33]. These densities of states represent the most fundamental spectra from ab initio computation and characterize the vibrational and electronic properties of materials. We use the phDOS dataset from ref. [34] and analyze the ability to extract thermodynamic properties from phDOS predictions. We use a computational eDOS dataset acquired from the Materials Project and focus on eDOS within 4 eV of band edges due to its importance for a breadth of materials properties. Given the computational expense of generating these spectra, the ability to predict these spectra, even in an approximate way, represents a valuable tool to perform materials screening to guide ab initio computation. We demonstrate such

acceleration of materials discovery with a use case based on the identification of band gaps below but near the Fermi energy in metallic systems, which have been shown to be pertinent to thermoelectrics and transparent conductors[35,36].

## Results

**Material-to-spectrum model architecture.** Mat2Spec is a model for predicting a spectrum (output labels) for a given crystalline material (input features) by: (i) encoding the labels and features onto a latent Gaussian mixture space, with the alignment of the feature and label embeddings to exploit underlying correlations; and (ii) contrastive learning to maximize agreement between the feature and label representations to learn a label-aware feature representation. These 2 modules can be considered as a multi-component encoder and decoder, respectively, that collectively form an end-to-end model whose architecture is detailed in Fig. 1.

Conceptually, Mat2Spec commences with a feature encoder whose high-level strategy is similar to that of E3NN and GATGNN, where each model aims to learn a materials representation that accurately captures how structure and composition relate to the properties being predicted. The first component of Mat2Spec's encoder is a graph neural network (GNN) that is based on previously-reported approaches to materials property prediction in which the GNN is the entirety of the encoder[9,25,26]. The remainder of the Mat2Spec framework introduces a new approach to learning from the GNN encodings and comprises our contribution to the design of machine learning models for materials property prediction. For deploying the trained model, we note that model inference only uses the feature encoder, representation translator, and predictor, where the feature encoder takes the input materials and produces probabilistic embeddings, the translator translates the probabilistic embeddings into deterministic representations, and the predictor reconstructs the final spectrum properties.

Compared to single-property prediction, spectrum prediction offers additional structure that may be captured through careful design of the model architecture. We consider discretized spectra where the input features that are important for the prediction of the intensity at one point in the spectrum are likely related to those of other points in the spectrum. While prior models such as

E3NN and GATGNN have no mechanism to explicitly encode these relationships, Mat2Spec captures this structure of the task with a probabilistic feature and label embedding generator built with multivariate Gaussians. During training, the generator operates on both the material (input features) and its spectrum (input label). In the process of label embedding, each point $i$ in the spectrum with dimension $L$ is embedded as a parameterized multivariate Gaussian $N_i$ with learned mixing coefficient $\alpha_i$, where $\sum_i \alpha_i = 1$. The spectrum for a material is thus embedded as a multivariate Gaussian mixture $\sum_i \alpha_i N_i$. The mixing coefficients capture relationships among the points in the spectrum where related points tend to have similar weights. To capture the common label structure among different materials, the set of multivariate Gaussians $\{N_i\}_{i=1}^L$ is shared across all materials.

For feature embedding from the GNN encoding of the material, we learn a set of $K$ multivariate Gaussians $\{M_j\}_{j=1}^K$ as well as a set of $K$ mixing coefficients $\{\beta_j\}_{j=1}^K$, and the features of a material is thus embedded as a multivariate Gaussian mixture $\sum_j \beta_j M_j$. Note that $K$ is a hyperparameter that is not required to be equal to the number of points in the spectrum. While the only input to the multivariate Gaussian for feature embedding is the output of the GNN, we leverage the learned structure of label embedding by promoting alignment of the two multivariate Gaussian mixtures, which both have dimension $D$, a hyperparameter. Specifically, the training loss includes the Kullback–Leibler (KL) divergence between the two multivariate Gaussian mixtures $\sum_i \alpha_i N_i$ and $\sum_j \beta_j M_j$. We want to note that, to reduce the computational overhead, all multivariate Gaussians have diagonal covariance matrices (which is similar to the variational autoencoders[37]). The label correlation is in fact modeled by the learned mixing coefficients. The number of Gaussians is equal to the number of points in each spectrum ($K'$), and the mixing coefficients capture relationships among the points in the spectrum where related points tend to have similar weights.

The probabilistic embedding generator provides the inputs to the contrastive learning decoder. The decoder commences with a shared multi-layer perceptron (MLP) Representation Translator, which during model training is evaluated once with the feature embedding to generate the feature representation and again with

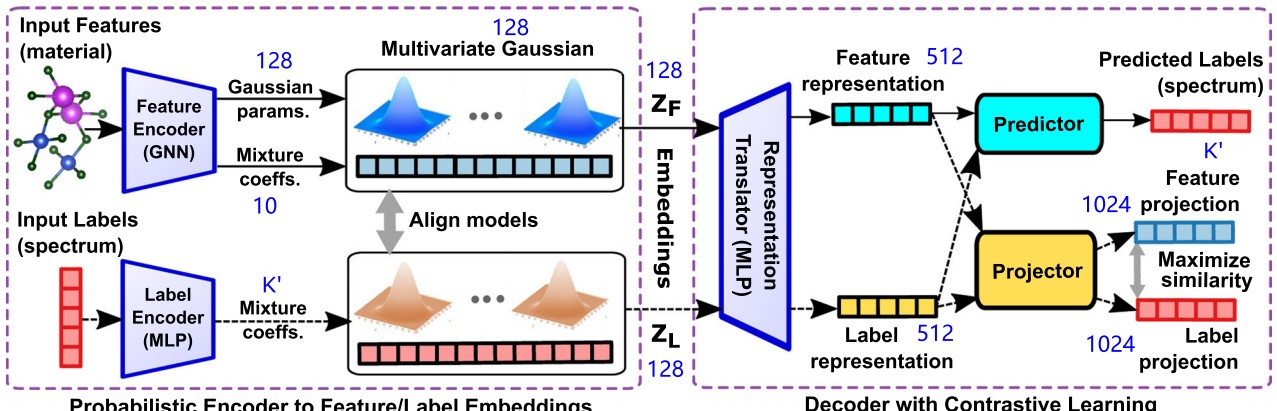

**Fig. 1 The Mat2Spec model architecture.** The prediction task of material (Input Features) to spectrum (Predicted Labels) proceeds with 2 primary modules, a probabilistic embedding generator (Encoder) to learn a suitable representation ($Z_F$) of the material and a Decoder trained via supervised contrastive learning to predict the spectrum from that embedding. The solid arrows show the flow of information for making predictions with the trained model, and the dashed arrows show the additional flow of information during model training, where the ground truth spectrum (Input Labels) is an additional input for which the Encoder produces the embedding $Z_L$. Alignment of the multivariate Gaussian mixture model parameters during training conditions the probabilistic generator of the embeddings. Both the input material (Feature projection) and input spectrum (Label projection) are reconstructed to train the model via contrastive learning. The Representation Translator is shared by the prediction and reconstruction tasks, resulting in parallel latent representations that are transformed into the final outputs by the Predictor and Projector, respectively. Note that the output dimension of each component is noted in blue, where $K'$ is 51 for phDOS and 128 for eDOS.

**Table 1 Results of phDOS prediction on the test set.**

| ML model | Setting | | phDOS prediction | | | | Calculated $C_V$ (300 K) | | Calculated $\bar{\omega}$ | |
|---|---|---|---|---|---|---|---|---|---|---|
| | Scaling | Loss | $R^2$ | MAE | MSE | WD | MAE | MSE | MAE | MSE |
| E3NN | MaxNorm | MSE | 0.56 | 0.094 | 0.034 | 39 | 3.58 | 56 | 30.9 | 3161 |
| GATGNN | MaxNorm | MSE | 0.45 | 0.105 | 0.042 | 44 | 4.66 | 80 | 35.1 | 3392 |
| Mat2Spec | MaxNorm | MSE | **0.63** | 0.086 | 0.029 | 33 | 3.30 | 49 | 26.2 | 2284 |
| E3NN | SumNorm | WD | −0.48 | 0.339 | 1.884 | 132 | 11.7 | 393 | 90.0 | 17343 |
| GATGNN | SumNorm | WD | −2.78 | 0.185 | 0.065 | 194 | 19.6 | 591 | 183 | 42753 |
| Mat2Spec | SumNorm | WD | 0.57 | 0.085 | 0.026 | **21** | **1.32** | **10** | **10.6** | **348** |
| E3NN | SumNorm | KL | 0.48 | 0.105 | 0.036 | 50 | 4.88 | 77 | 41.1 | 3718 |
| GATGNN | SumNorm | KL | −1.05 | 0.177 | 0.057 | 215 | 22.4 | 756 | 205 | 51609 |
| Mat2Spec | SumNorm | KL | 0.62 | **0.078** | **0.023** | 24 | 1.96 | 11 | 17.1 | 625 |

For each combination of 3 ML models and 3 settings, the performance metrics include 4 measures of the prediction of the 51-D phDOS and 2 measures each for the properties CV (heat capacity at 300 K, J/K · mol) at 300K and $\bar{\omega}$ (average phonon frequeny, cm$^{-1}$) calculated from each material's phDOS. Each loss metric is aggregated over all materials in the test set. Note that for phDOS predictions with MaxNorm scaling, each prediction is re-scaled to a maximum value of 1 prior to evaluating prediction loss, and analogously the predictions with SumNorm scaling are re-scaled to a sum of 1 prior to evaluating prediction loss. MAE and MSE metrics for phDOS prediction inherit the arbitrary units from the respective scaling and should not be directly compared across scaling. $R^2$ (unitless) and WD (units of cm$^{-1}$ from the energy axis) are insensitive to this difference in scaling. The best value in each column is noted in bold. In each setting and for each loss metric, Mat2Spec provides the best predictions.

the label embedding to generate the label representation. Using these representations, the Predictor and Projector are shallow MLP models that produce the predicted spectrum and feature/label projections, respectively. While only the Predictor is used when making new predictions, the Projector maps representations to the space where the contrastive loss is applied during training. The Representation Translator and Projector are trained to maximize agreement between representations using a contrastive loss, making the feature representation label-aware. We follow the empirical evidence that the contrastive loss is best defined on the projection space rather than the representation space[38].

For spectral data that is akin to a probability distribution for each material, the prediction task has additional structure, for example, that an increased intensity in one portion of the spectrum predisposes other portions of the spectrum to have lower intensity. The Mat2Spec strategy of learning relationships among how input features maps to multiple labels supports this aspect of distribution prediction, although deployment of Mat2Spec for distribution learning is ultimately achieved by training using either the Wasserstein distance (WD) or KL loss, which each quantify the difference in 2 probability distributions, as the primary training loss function. Combined with more traditional loss functions, we compare Mat2Spec with baseline models E3NN and GATGNN with three different settings, i.e., different combinations of data scaling and training loss function. Note that our adaptions of E3NN and GATGNN for distribution-based training constitute models beyond those previously reported that are referred to herein by the respective name of the reported model architecture.

**Predicting phonon density of states.** The task of phDOS prediction of crystalline materials was recently addressed by E3NN[14]. To provide the most direct comparison to that work, we commence with their setting wherein each phDOS is scaled to a maximum value of 1 (MaxNorm), and mean squared error (MSE) loss function is used for model training. An important attribute of phDOS is that the integral of the phDOS can be approximated as 3 times the number of sites in the unit cell. As a result, not only are data scaled for model training, but the output of a ML model can be similarly scaled as the final step in spectrum prediction. Since our ground truth knowledge about scaling is with regards to the sum of the phDOS as opposed to the maximum value, we also consider scaling data by its sum (NormSum). This scaling make

phDOS mathematically equivalent to a probability distribution, motivating incorporation of ML models for learning distributions, i.e., predicting the distribution of phonon energies in a material with a known number of phonons. Our 2 corresponding settings for phDOS prediction use NormSum scaling paired with each of WD and KL loss functions. The combination of 3 ML models and 3 settings provides 9 distinct models for phDOS prediction whose performance is summarized in Table 1 using 4 complementary prediction metrics: the coefficient of determination ($R^2$ score, which is typically between 0 and 1 but can be negative when the sum of squares of the model's prediction residuals are larger than the total sum of squares of difference of the observation values from their mean), mean absolute error (MAE), MSE, and WD. In each of the 3 settings, Mat2Spec outperforms E3NN and GATGNN for all 4 of these metrics. The results are most comparable among the 3 ML models in the MaxNorm-MSE setting, and the best value for each metric is obtained with one of the Mat2Spec models.

To highlight the value of phDOS prediction, Chen et al.[14] noted the importance of calculating properties such as the average phonon frequency $\bar{\omega}$ and the heat capacity $C_V$ at 300 K. Table 1 shows the MAE and MSE loss for these quantities calculated from the test set of each of the 9 phDOS prediction models. Mat2Spec in the SumNorm-WD setting provides the best performance for both metrics and for both physical quantities. Figure 2 summarizes the range of prediction errors for these quantities, demonstrating that in each setting, Mat2Spec is not only optimal in the aggregate metrics but also has a lower incidence of extreme outliers, further highlighting how this model facilitates generalization to accurate spectral prediction for all materials.

**Predicting electronic density of states.** We consider the prediction of the total electronic DOS (eDOS) of nonmagnetic materials, which is more complex than phDOS prediction in a variety of ways. While a full-energy density of occupied states would share the phDOS attribute of having a known sum for each material due to the known electron count of the constituent elements, the eDOS is of greatest interest in a relatively small energy range near the band edges and including both unoccupied and occupied states. In the present work, we consider an energy grid of 128 points spanning −4 to 4 eV with respect to band edges with 63 meV intervals. On this energy grid, the Fermi energy, as well as the band edges where applicable, are all set to 0 eV; in what follow, we remove the zero-valued eDOS region between the

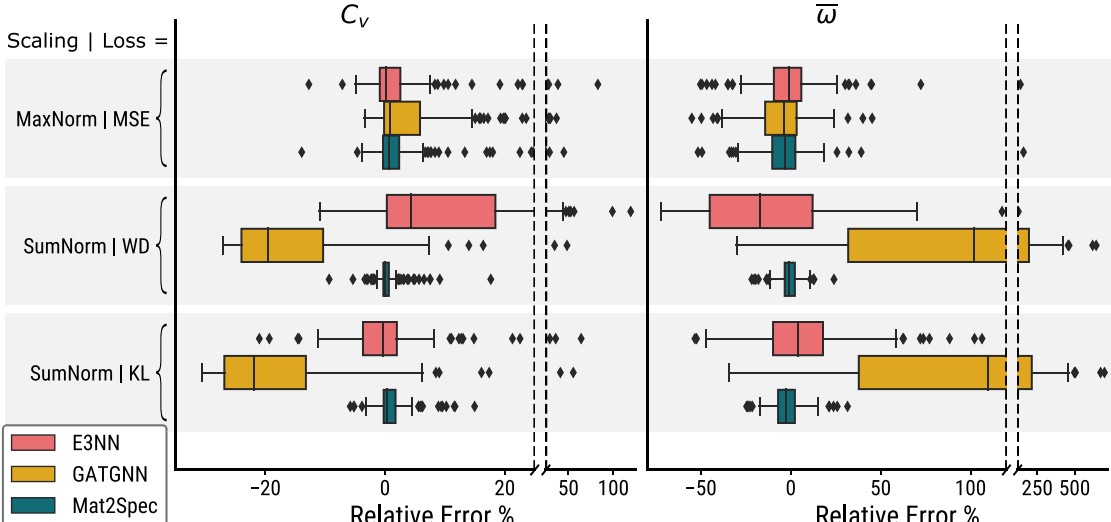

**Fig. 2 Comparison of metrics calculated from phDOS in the test set.** For each of 3 settings (horizontal shaded bars), the results of the $C_V$ (heat capacity at 300 K, left) and $\bar{\omega}$ (average phonon frequency, right) calculations from the predicted phDOS in the test set are shown as the relative error with respect to the ground truth value for each material in the test set. For each of the 3 settings, 3 models, and 2 properties, the relative errors are shown with a box plot (center line, median; box limits, upper and lower quartiles; whiskers, 1.5× interquartile range; points, outliers). In the MaxNorm-MSE setting, all 3 ML models have similar median relative losses for both $C_V$ and $\bar{\omega}$, with Mat2Spec providing a smaller interquartile range and less extreme outliers. In the other settings, Mat2Spec outperforms the other ML models with respect to median, interquartile range, and outliers.

valence band maximum (VBM) and conduction band minimum (CBM) for materials with a finite bandgap. While predicting the bandgap energy is also important, this task is the subject of major effort in the community[11,18,19], and we focus here on the prediction of eDOS surrounding the band edges of greatest value.

As a traditional setting for eDOS prediction, we use zero-mean, unit-variance (Standard) scaling for each energy point in the 128-D grid with MAE training loss. The SumNorm scaling with both WD and KL loss remain important settings, and when each ML employs one of these distribution-learning settings, the predictions are re-scaled to the native eDOS units of states/eV using the sum of the prediction from the Standard-MAE setting. As a result, each ML model in each setting produces eDOS prediction in the native units such that the metrics for eDOS prediction can be compared across the set of 9 eDOS prediction results. Note that per its definition, the computational of the WD metric involves SumNorm scaling of all ground truth and predicted eDOS.

Figure 3 shows representative examples of phDOS and eDOS ground truth and predictions, with additional examples provided in Supplementary Figs. 1 and 2. For each ML model, the test set materials are split into 5 equal-sized groups based on the quintiles of MAE loss. For each quintile, the intersection over the 3 ML models provides a set of materials with comparable relative prediction quality. Selection from each quintile set enables visualization of good (left) to poor (right) predictions. In phDOS prediction, models can take advantage of the broad range of materials with similar phDOS patterns, as evidenced in the lower quintiles of Fig. 3 and Supplementary Fig. 1. The eDOS, particularly with an appreciable energy window, exhibits more high-frequency features with fewer analogous characteristic patterns. For both phDOS and eDOS, Mat2Spec consistently predicts the correct general shape of each pattern with regression errors arising mostly from the imperfect prediction of the magnitude of sharp peaks.

Table 2 provides the eDOS performance metrics from each combination of 3 ML models and 3 settings using the same 4 regression metrics as shown for phDOS prediction. Here, the WD metric is calculated using SumNorm scaling and thus measures

the quality of distribution prediction under an assumption that the sum of each material's eDOS is known. While the best performance against the WD metric in phDOS prediction was achieved using WD as training loss, Mat2Spec in the SumNorm-KL setting provides the best eDOS results with respect to the WD, $R^2$, and MSE metrics. We anticipate that model training with the WD loss function is compromised by the many sharp peaks in eDOS data, although a full diagnosis of the relatively poor performance in eDOS compared to phDOS prediction for the SumNorm-WD setting may be pursued in future work.

The SumNorm-KL setting improves the eDOS prediction for each ML model compared to the Standard-MAE setting for nearly all of the prediction metrics, with the only exception being a slight increase in MAE for Mat2Spec. While these results highlight the general value of using distribution learning with any ML model, the superior performance of Mat2Spec with respect to the baseline models in the SumNorm-KL setting highlights the particular advantages of distribution learning when using probabilistic embedding generation and contrastive learning to condition the model.

In the Supplementary Figures, we explore the underpinnings of the distribution in prediction accuracy of Mat2Spec in the SumNorm-KL setting. Intuitively, the prediction task is harder and the MAE is generally larger when (i) the material's structure is more complicated, (ii) there are few examples of similar materials in the training set, and/or (iii) the eDOS contains high frequency features, which are all demonstrated in Supplementary Figs. 3, 4, 5, and 6. Supplementary Figure 5 illustrates that the distribution of prediction quality can vary substantially with materials class, which is likely due to a convolution of the aforementioned effects with the chemical complexity of a given subclass of materials.

The probabilistic embedding generator, as well as the contrastive learning, are intended to amplify knowledge extraction from data. One way to assess success toward this goal is to study the prediction performance with decreasing training size, as shown in Fig. 4. For the complementary prediction loss metrics MAE and WD, Mat2Spec has relatively little degradation in

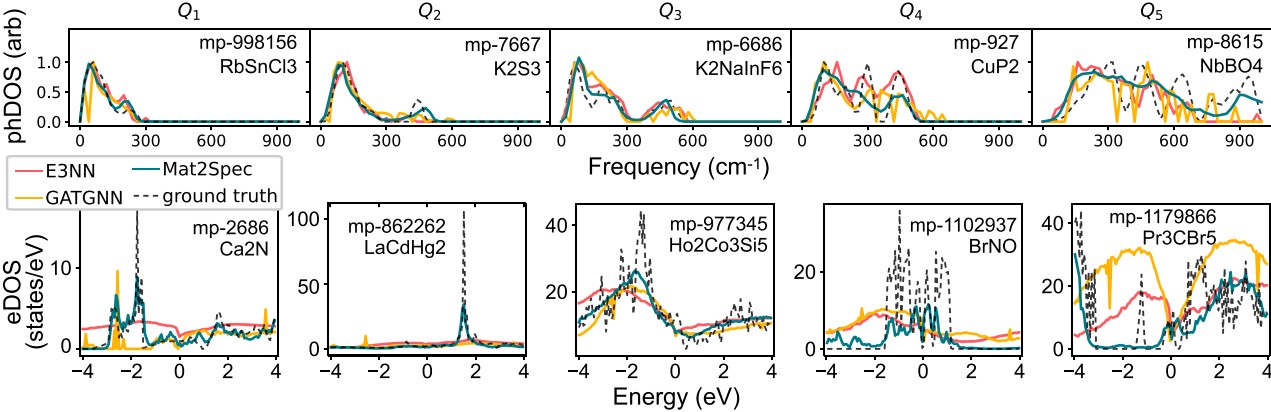

**Fig. 3 Example phDOS and eDOS predictions in the test set.** Ground truth and ML predictions are shown for five representative materials chosen from the 5 quintiles from low MAE loss ($Q_1$, left) to high MAE loss ($Q_5$, right) for both phDOS (top) and eDOS (bottom). The best overall setting for each model is used, which is SumNorm-KL except for phDOS prediction by E3NN and GATGNN that use MaxNorm-MSE. From $Q_1$ to $Q_5$ the ground truth phDOS is increasingly complex. For $Q_1$ to $Q_3$, Mat2Spec predicts each phDOS feature, where the other models are less consistent. In $Q_4$, E3NN and Mat2Spec are comparable with their prediction of three primary peaks in the phDOS. In $Q_5$, the presence of substantial density above 600 cm$^{-1}$ is quite rare, and Mat2Spec is the only model to make the correct qualitative prediction. For eDOS, there is no analogous change in the shape of the eDOS across the quintiles. In $Q_1$ and $Q_2$, Mat2Spec provides the only qualitatively correct predictions. In $Q_3$, each model predicts a smoothed version of the ground truth. In $Q_4$, Mat2Spec prediction is far from perfect but is the only prediction to capture the series of localized states near the Fermi energy. In $Q_5$, each model has qualitatively comparable predictions in the conduction band, but Mat2Spec is the only model to capture the primary structure of the valence band. Mat2Spec's under-prediction of one localized state makes this one of its highest MAE predictions, which is far lower than the worst predictions from other models. The ability of Mat2Spec to globally capture the qualitative patterns for both phDOS and eDOS leads to its superior performance for each metric in Tables 1 and 2.

**Table 2 Results of eDOS prediction.**

| ML model | Setting | | eDOS prediction | | | | VB gap identification | | | VB gap discovery | |
|---|---|---|---|---|---|---|---|---|---|---|---|
| | Scaling | Loss | $R^2$ | MAE | MSE | WD | F1 | Precision | Recall | Precision | Recall |
| E3NN | Standard | MAE | 0.39 | 5.24 | 105.1 | *0.48* | 0.035 | 0.333 | 0.019 | – | 0 |
| GATGNN | Standard | MAE | 0.30 | 4.89 | 120.9 | 0.42 | 0.182 | 0.263 | 0.139 | 0 | 0 |
| Mat2Spec | Standard | MAE | 0.53 | **3.64** | 80.4 | 0.27 | 0.352 | 0.509 | 0.269 | 0.67 | 0.27 |
| E3NN | SumNorm | WD | *−2.1* | *9.81* | *542.8* | 0.40 | *0* | *0* | *0* | – | 0 |
| GATGNN | SumNorm | WD | 0.19 | 6.41 | 140.1 | 0.26 | 0.018 | 0.200 | 0.009 | 0 | 0 |
| Mat2Spec | SumNorm | WD | 0.38 | 5.23 | 107.7 | 0.29 | 0.018 | 0.250 | 0.009 | – | 0 |
| E3NN | SumNorm | KL | 0.41 | 5.01 | 101.9 | 0.47 | *0* | *0* | *0* | – | 0 |
| GATGNN | SumNorm | KL | 0.32 | 4.89 | 118.2 | 0.35 | 0.243 | 0.450 | 0.167 | 0.13 | 0.20 |
| Mat2Spec | SumNorm | KL | **0.57** | 3.8 | **74.5** | **0.21** | **0.397** | **0.698** | **0.278** | 0.47 | 1.00 |

For each combination of 3 ML models and 3 settings, the performance metrics include 4 measures of the prediction of the full 128-D eDOS (evaluated on the test set), 3 measures of the classification accuracy for identification of gaps in the eDOS in the VB (evaluated on the test set), and 2 measures of the classification accuracy for 100 materials with no prior available eDOS. Note that the predictions from SumNorm scaling are re-scaled to the original eDOS units using the normalization factor provided by the Standard scaling prediction from the respective ML model. As a result, MAE loss is in native eDOS units of states/eV, and MSE has the square of these units; WD has units of eV from the energy axis, and the remainder of the metrics are unitless. For the metrics calculated on the test set, the best value is in bold and worst value is in italics, and Mat2Spec exhibits the best value for each metric. The 100 materials for which we performed DFT calculations include 49 random selections as well as predicted positives from GATGNN and Mat2Spec in the SumNorm-KL setting. Precision values are missing for 4 models with no positive predictions among the 100 materials.

performance with decreasing training data, providing better predictions when using 1/8 of the training data than the baseline models that use the full training data. The ability to make meaningful predictions with data sizes on the order of 10$^3$ materials is critical for predicting spectral properties that are much more expensive to acquire, where the increased expense typically corresponds to less available training data than the eDOS dataset used in the present work. The increased expense also expands the opportunity for accelerating materials discovery with Mat2Spec.

The excellent performance with relatively small training size can also be transformative for materials discovery campaigns conducted within a specific subclass of materials. Using 2 example subclasses from Supplementary Fig. 5, 3-element oxides without rare earths and materials with spacegroup 225, we show that Mat2Spec can be fine-tuned to improve the predictions

within each subclass (see Supplementary Fig. 8), which enables Mat2Spec to more accurately model specific eDOS features, as shown in Supplementary Fig. 9. Even when ones research is confined to a specific subclass, training Mat2Spec on a broader set of materials helps Mat2Spec encode materials chemistry, as demonstrated by a substantial degradation in performance when using only the data from a specific subclass (see Supplementary Fig. 8).

While our focus on phDOS and eDOS prediction is precisely motivated by their transcendence of specific uses of materials, evaluating eDOS prediction for a specific use case provides a complementary mechanism for gauging the quality and value of the predictions. For a use case, we choose a classification of materials based on a desirable feature in the eDOS for certain materials. We consider the presence of energy gaps (or near-energy gaps) in occupied states close to the Fermi energy, which

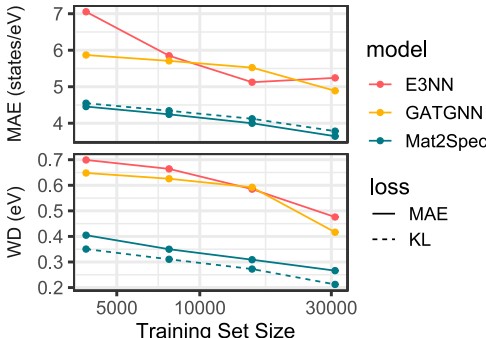

**Fig. 4 Data size dependence of eDOS prediction.** Using a static test set, random down-selection of the train set by factors of 2, 4, and 8 enables characterization of how the prediction loss varies with size of the training set. This study was performed for each ML model in the Standard-MAE setting as well as for Mat2Spec in the SumNorm-KL setting. As expected, prediction loss generally increases with decreasing training size. Using only 1/8 of the training data, Mat2Spec (in either setting) provides lower MAE and WD losses compared to the baseline models that use the full training set. Achieving better results with an 8-fold reduction in data size highlights how the structure of Mat2Spec conditions the model to learn more with less data.

corresponds to an intrinsic eDOS that is similar to that of a degenerate semiconductor. Correspondingly, the metals exhibiting an energy gap in occupied states may exhibit transport-related properties such as electronic conductivity and a Seebeck coefficient that mimic a doped semiconductor[35], broadening the materials search space. Indeed, this concept of metallic system having a gap close to the Fermi energy was introduced and studied in the research of potential transparent conductive materials[36,39–41], for applications in low-loss plasmonics[42], and in catalysis[43].

Recently, a high-throughput search for "gapped metals" exploited this key feature of the eDOS to find new potential thermoelectric materials[35]. For the present use case we focus on materials with a single energy gap below but near the Fermi energy, hereafter called the "VB gap", and we evaluate the ability to recognize this feature in a broad range of materials, considering the entire test set (containing metals and semiconductors). We note that a similar approach can be applied for gaps in the conduction band. For a VB gap to be sufficiently interesting to merit follow-up study of the material, (i) the eDOS must be sufficiently low in intensity throughout the VB gap, for which we use a threshold of 1 state/eV; and (ii) the energy gap must be sufficiently wide in energy, for which we use a 1 eV minimum gap width, (iii) sufficiently close to the VB edge, for which we require that the high-energy edge of the gap is no smaller than −1 eV, and (iv) sufficiently far the Fermi energy so that there are available carriers, for which we require that the high-energy edge of the gap is no larger than −0.25 eV. This stringent set of criteria corresponds to a low positivity rate, in particular only 2.8% of materials in the test set meet all four criteria, highlighting the inherent challenge of discovering such materials.

To ascertain the ability of each prediction model to identify such materials, the binary classification for the presence of a VB gap was evaluated, as summarized by the precision, recall, and F1 scores in Table 2. The requirement for eDOS to remain below the threshold for a wide, contiguous energy range requires the predictions to be simultaneously accurate for many energies, which is aligned with our strategy for distribution-based learning. At the same time, any over-prediction of eDOS in the relevant energy range disrupts the detection of a VB gap, and since KL loss penalizes errors in small eDOS values to a greater extent than

WD, the SumNorm-KL setting is intuitively the best approach for this use case, which is reflected in its improved classification scores for both the GATGNN and Mat2Spec models compared to other settings. Each model in the SumNorm-WD setting, as well as E3NN in all settings, provide poor performance for this classification task, highlighting that this is an aggressive use case that is emblematic of materials discovery efforts wherein one seeks a select set of materials exhibiting unique properties. In order to highlight the importance of each part of the model in achieving the best accuracy, we also performed ablation studies in which (i) the label probabilistic embedding generator was removed to eliminate model alignment during Mat2Spec training, and (ii) the projector was removed to eliminate the supervised contrastive learning. Therefore, the model alignment loss and supervised contrastive loss were also removed, respectively. We performed ablation studies with the eDOS prediction and SumNorm-KL setting. The resulting MAEs are 4.05 and 4.20, which are 6.6 and 10.5% higher, respectively, than that of the original Mat2Spec with the SumNorm-KL setting. The resulting WDs are 0.24 and 0.27, which are 14.3 and 28.6% higher, respectively, than that of the original Mat2Spec with the SumNorm-KL setting, demonstrating that removing these key components of Mat2Spec substantially degrades performance.

Figure 5 summarizes, for both eDOS prediction and the VB gap use case, the relative performance across the 3 ML models and 3 settings using radar plots in which each axis is scaled by the minimum and maximum value (or vice versa for loss metrics where lower is better) observed over the 9 prediction models. An eDOS model that is accurate with respect to each regression metric from eDOS prediction, as well as the use case classification scores, will appear as the largest shaded region in the radar plot. In the Standard-MAE setting, Mat2Spec clearly outperforms the other models, and its performance is further enhanced in the SumNorm-KL setting.

**Guiding discovery of materials with tailored electronic properties.** To further demonstrate the importance of eDOS prediction for materials discovery, we extend the VB gap use case to the set of nonmagnetic materials in the Materials Project for which no eDOS has been calculated. The primary computational screening for materials with specific properties includes the evaluation of performance-related properties as well as basic properties of the materials. In addition to the VB gap requirements, we consider materials that are relatively near the hull while also considering enough materials to generate meaningful results, prompting an upper limit of 0.5 eV/atom of the free energy hull. For simplicity, we only consider materials with fewer than five elements. To make the search specific to gapped metals, we also require the band gap to be zero, which for materials with no available eDOS corresponds to the estimated band gap from the Materials Project structural relaxation calculation.

Querying the Materials Project for materials that lack eDOS and meet the energy above hull, number of elements, and band gap requirements produced 8,106 candidate materials. DFT calculations were performed on 100 of these materials, including 49 randomly-selected materials as well as the 51 materials predicted to have a VB gap by either the Mat2Spec SumNorm-KL or GATGNN SumNorm-KL models. Table 2 shows the excellent performance of Mat2Spec with precision and recall of 0.47 and 1.0, respectively, both exceeding the GATGNN values of 0.13 and 0.2, respectively. There are 4 materials for which both Mat2Spec and GATGNN predict a VB gap, with 3 of these validated as TP by DFT. There are 19 materials predicted positive by GATGNN but negative by Mat2Spec, and none of these were found to have a VB gap via DFT, i.e., Mat2Spec correctly predicted all of these as

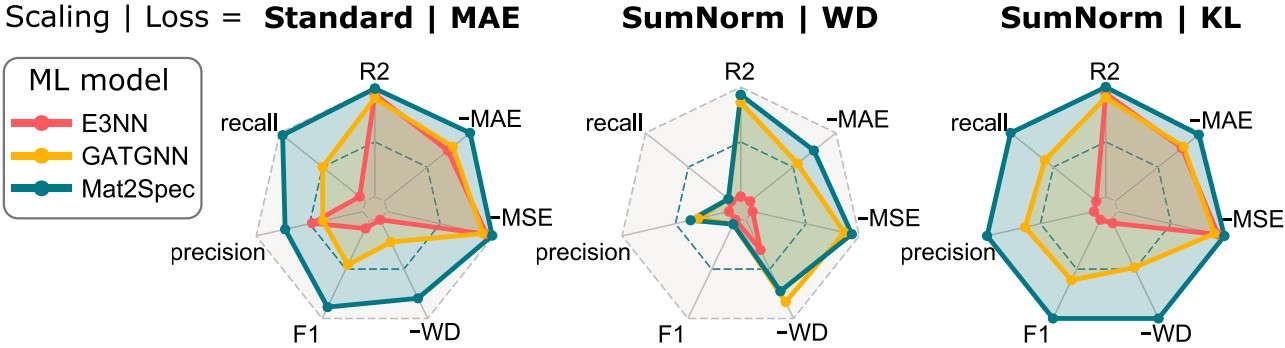

**Fig. 5 Summary of loss metrics for eDOS prediction in the test set.** The three panels correspond to 3 eDOS settings, and each shows results for the 3 ML models for a total of 9 radar plots corresponding to the 9 rows in Table 2. In the table, the worst and best values are shown in italics and bold, respectively, providing the minimum and maximum values for each axis of the radar plots so that maximum value on each axis corresponds to the best performance. Since each axis uses the same scaling, all 9 radar plots are directly comparable. The axes include 4 metrics for 128-D eDOS prediction ($R^2$, MAE, MSE, WD) and 3 metrics for the classification of gaps in the VB. For each metric, Mat2Spec provides the best value in either the Standard-MAE or SumNorm-KL settings, with Mat2Spec predictions in the SumNorm-KL setting providing the best collective performance as visualized by its maximal area among the radar plots.

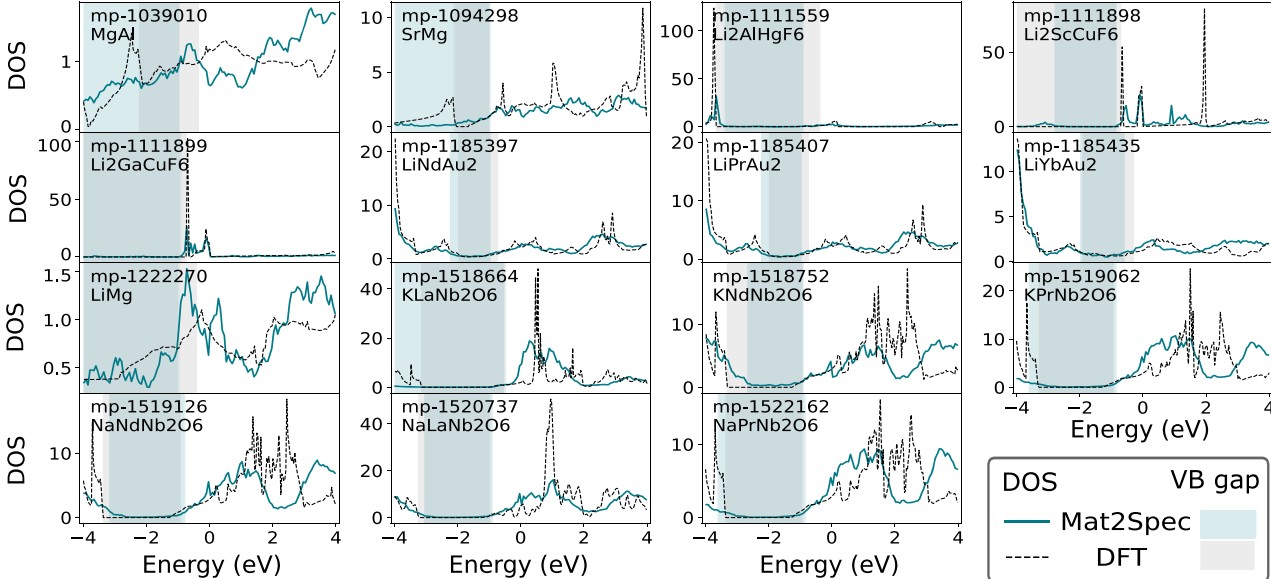

**Fig. 6 Discovered materials with VB gaps.** Of the 32 predictions of materials with a VB gap according to the results of the Mat2Spec eDOS predictions in the SumNorm-KL setting, the 15 TP predictions are shown. For each material, the eDOS calculated from DFT is shown with the Mat2Spec prediction, and the energy range of the VB gap is also shown. MgAl and LiMg have a VB gap but the inherently low electron density of the material and the persistence of a small but finite eDOS to below -3 eV limits the interest in these materials. LiRAu$_2$ (where R = Nd, Pr, and Yb) have low eDOS considering the presence of heavy elements, although the eDOS does not reach zero in the energy range of interest. Zero density in the energy range of interest is observed in the 10 other materials that fall within 2 families of candidate thermoelectrics. In the family of fluorides with formula Li$_2$AMF$_6$ (where A = Al, Sc, Ga; M = Hg, Cu), each material exhibits a gap of 3–4 eV starting near 0.5 eV below Fermi energy. The eDOS of each material in this family also exhibits a near-gap above the Fermi energy, which motivates their further study for applications such as transparent conductors. A family of oxides with formula ARNb$_2$O$_6$ (where A = Na, K; R = La, Nd, Pr) share a similar eDOS with a VB gap that is 2–3 eV wide starting near 1 eV below Fermi energy.

negatives. Of the 28 materials predicted positive by Mat2Spec but negative by GATGNN, 12 were found to have a VB gap via DFT, i.e., only Mat2Spec enabled discovery of these 12 gapped metals. Of the randomly selected materials, none were found to have a VB gap, which is commensurate with the expectation that the positivity rate in the set of candidate materials is similarly low as that of the test set. Collectively the results show the excellent performance of Mat2Spec for the discovery of these rare materials, where 15 discoveries were validated by DFT from Mat2Spec's predicted positives. This down-selection by a factor of 253 from the set of candidates corresponds to a proportional savings in the computation time for discovering gapped metals.

Figure 6 shows the DFT calculation and Mat2Spec prediction for each of the 15 TP materials, revealing excellent qualitative agreement and leading to the identification of a family of fluorides Li$_2$AMF$_6$ (A = Al, Sc, Ga; M = Hg, Cu) and a family of oxides ARNb$_2$O$_6$ (A = Na, K; R = La, Nd, and Pr) that are of particular interest. To assess these materials as potential thermoelectrics, we note that thermoelectric candidates should maximize the power factor (PF), PF = $S^2\sigma$, where $S$ is the Seebeck coefficient and $\sigma$ is the electrical conductivity. According to Mott's formula[44] a rapid variation of the eDOS would increase the Seebeck coefficient; since the presence of a band gap close to the Fermi energy makes the DOS sharply decrease, the Seebeck

coefficient increases significantly and the PF presents a distinct peak in this case.

Known high-PF thermoelectric are indeed gapped metals, such as La$_3$Te$_4$, Mo$_3$Sb$_7$, Yb$_{14}$MnSb$_{11}$, and NbCoSb, along with others[35], and present an eDOS similar to those shown in Figure 6. This similarity allows us to use the presence of the gap combined with the fast rise of the eDOS as a means for identifying systems with large Seebeck coefficients. In particular, looking at Fig. 6, the flourides possess both characteristics. Also, a high carrier concentration ($\sim10^{23}$ cm$^{-1}$) associated with the partially filled bands between the gap and the Fermi energy may lead to high electrical conductivity. In contrast, for oxides, since the rise of the eDOS is less rapid than in the flourides, the Seebeck coefficient is expected to be lower.

Another potential use of materials with large VB gaps would be as transparent conductors. As defined in refs. [36,39] (i) metals with large VB gaps can reduce the interband absorption in the visible range; if they also possess (ii) a high enough carrier density in the CB to provide conductivity, and (iii) sufficiently low carrier density to limit the interband transition in the CB and plasma frequency ($\omega_p \sim \sqrt{n_e/m}$, where $m$ is an effective carrier mass), free-electron absorption will not interfere with the needed optical transparency. Inspecting the eDOS of the flourides and oxides suggested by our models, we observe that they firmly meet criterion (i) with eDOS similar to that of the materials investigated in ref. [36] such as Ba–Nb–O and Ca–A–O systems. Regarding criterion (ii), as mentioned before, the fluorides and oxides in Fig. 6 have a carrier density of approximately $10^{23}$ cm$^{-1}$, which is comparable to that of known intrinsic transparent conductors. The assessment of the plasma frequency for criterion (iii) requires an estimation of the carrier effective mass, which is beyond the scope of this work. Also, the flourides have a eDOS close to those referred to as Type-1 transparent conductors in ref. [39] that are metals with the Fermi energy located in an intermediate band which is energetically isolated from the bands below and above it.

While these qualitative assessments require a quantitative validation via detailed computation in future work, the fact that the eDOS compare well to established literature for thermoelectric and transparent conductor materials strongly motivates future detailed studies of these systems. Since the oxide family includes transition metals and the fluoride family includes rare earth elements, initial steps could include validation of the position of the d-electron and f-electron states, respectively, as well as the absence of a band gap with techniques more accurate than standard DFT. Future assessment of the electronic and thermal transport properties can confirm their potential as thermoelectrics, and optical properties including the dielectric function can be used to gauge their suitability as transparent conductors. The most promising materials from these computational assessments would then be prime candidates for assessing synthesizability, where the experimentally realized materials can then be analyzed to further validate the computed properties.

## Discussion

Mat2Spec brings together several machine learning techniques to exploit prior knowledge and problem structure and to compensate for the relatively small amount of training data compared to the breadth of possible materials and DOS patterns. Specifically, Mat2Spec's feature encoder was inspired by the GATGNN[9], CGCNN[26], and MEGNet[12] models in which GNNs are used to encode the crystal structures. The encoding of the labels and features onto a latent Gaussian mixture space to exploit underlying correlations was inspired by the DMVP[45,46], DHN[47], and H-CLMP[7] models in which the latent Gaussian spaces learn multiple properties' correlations. Integrating correlation learning

with neural networks was initially motivated by computational sustainability applications[48]. The learning of a label-aware feature representation with contrastive learning was inspired by the SimSiam[49] model in which contrastive learning is used to maximize mutual information between two latent representations. Generalization of ML models is facilitated by incorporating techniques from disparate domains that share commonalities in the types of relationships that need to be harnessed by the models.

Our design of Mat2Spec focused on encoding label and feature relationships and how composition and structure give rise to intensity in different portions of a given type of spectrum. Interpretability of ML models remains a key challenge for advancing the fundamental understanding, and the information-rich embeddings generated by the Mat2Spec encoder motivate future work in analyzing them and their probabilistic generator to reveal the materials features that give rise to specific spectral properties. Such studies are increasingly fruitful with improved prediction capabilities[50], furthering the importance of Mat2Spec's improved performance compared to baseline models. The Mat2Spec prediction capabilities are further emphasized by the demonstrated identification of relevant and rare features, such as the presence of a VB gap in the eDOS, where an initial down-selection of candidate materials is particularly impactful for accelerating discovery. Our demonstration of discovering gapped metals can be extended in future work by coupling to crystal structure generators[51,52] to predict the eDOS and the phDOS of thousands of hypothetical compounds at a computational time that is orders of magnitude smaller than performing standard DFT computations. In Supplementary Figure 7 we show that for many materials, comparable eDOS predictions can be made using unrelaxed structures, indicating that Mat2Spec may be deployed for predicting the eDOS of any hypothetical material provided the atomic coordinates of each generated structure are sufficiently similar to those of a DFT-relaxed structure to capture the materials chemistry. This mode of deployment of Mat2Spec motivates the future study of the sensitivity of eDOS to atomic coordinates and lattice parameters for both DFT-calculated and Mat2Spec-predicted eDOS. Our results also indicate that Mat2-Spec can provide greater computational savings when applied to small but higher quality spectral datasets, which are typically more expensive to compute ab initio or to measure experimentally. Mat2Spec can also be extended to different energy ranges and energy resolutions, or even to spectra obtained from formalisms beyond DFT, as required for a given materials discovery effort.

Our focus on predicting fundamental spectral properties of materials is meant to demonstrate the generality of the Mat2Spec framework. Demonstration of the model's learning of appropriate materials embeddings for prediction of both phDOS and eDOS indicates that the model will also be suitable for spectral properties calculated from the phDOS or eDOS, such as phonon scattering and spectral absorption. We hope that the community will participate in the extension of Mat2Spec to these other domains of materials property prediction. We also want to note that our model architecture and distribution-based learning are applicable to scientific domains beyond materials science. Mat2-Spec's probabilistic embedding generator affords significant control over how we model our latent distribution via a prior distribution. Therefore, our model can be generalized to other domains by choosing appropriated prior distributions, model alignment losses, and prediction losses. For example, we can choose Poisson loss for counting problems, such as the species' abundance prediction[47], and binary cross entropy loss for classification problems, such as the species' present and absent prediction[45]. As a consequence of these underlying connections

among different domains, the Mat2Spec architecture can address a broader family of problems within and beyond materials science, an exemplar of our recently-described opportunity to exploit computational synergies between materials science and other scientific domains[53].

## Methods

**Mat2Spec model**. The pseudocode of Mat2Spec is in Table 3. Given input features $F$ and its corresponding input labels $L$, Mat2Spec first embeds them into two latent embeddings $Z_F$ and $Z_L$ via the feature and label encoders, respectively. These two embeddings are designed to exploit feature and label correlations implicitly. This is inspired by the label embedding technique[54] where both instances and their labels are encoded onto a structured latent space to align the instance embedding with the corresponding label embedding. While the traditional label embedding techniques assume a deterministic latent space, which lacks feature and label representation smoothness and thus increases sensitivity to input noise, Mat2Spec learns a probabilistic latent space where we can have significant control over the design of the latent space via a prior distribution.

*Crystal structure feature encoder and label encoder.* For an input crystal structure, we have not only element composition but also spatial information about the atoms. To leverage the spatial information, each element in the input is represented by an initial unique vector. These initial unique element embeddings capture some prior knowledge about correlations between elements[26,55]. These initial representations are then multiplied by a $N$ by $M$ learnable weight matrix where $N = 92$ is the size of the initial vector and $M = 128$ is the size of the internal representations of elements used in the model. The graph attention networks (GATs)[56] is then used to update these initial internal representations by propagating contextual information about the different elements present in the material between the nodes in the graph. GATs are constructed by stacking a number of graph attention layers in which nodes are able to attend over their neighborhoods' features via the self-attention mechanism. Specifically for each atom $i$, we first get its set of neighbors $\mathcal{M}_i = \{j | d(i, j) \leq \gamma\}$ where $d(i, j)$ denotes the Euclidean distance between $i$ and $j$ in angstroms and $\gamma = 8 \in \mathbb{R}^+$ is a predefined threshold. Then for each $j \in \mathcal{M}_i$, the distance $d(i, j)$ is encoded as a vector $u_{ij} = \langle f(T[0]), \cdots, f(T[m]) \rangle$ where $T \in \mathbb{N}^m$ is any predefined vector, $T[k]$ is the $k$-th element of $T$, and $f$ is the Gaussian probability density function:

$$f(x) = \frac{1}{\sigma\sqrt{2\pi}} \exp\left(-\frac{1}{2}\left(\frac{x - d(i,j)}{\sigma}\right)^2\right), \quad (1)$$

where $\sigma = 0.2$ is a predefined standard deviation parameter.

Furthermore, we use the element composition vector as a global context vector characterizing the entire crystal graph and concatenate the global context vector with each node feature vector and its corresponding edge feature vector. Therefore, the attention coefficient between every pair of neighbor nodes is updated as

$$e_{ij} = a(\mathbf{W}(h_i \oplus u_{ij} \oplus c), \mathbf{W}(h_j \oplus u_{ij} \oplus c)), \quad (2)$$

where $\oplus$ denotes the concatenation operation, $h_i, h_j \in \mathbb{R}^d$ are node features, $u_{ij} \in \mathbb{R}^m$ denotes an edge feature, $c \in \mathbb{R}^n$ denotes element composition, $\mathbf{W} \in \mathbb{R}^{d \times (d+m+n)}$ is a weight matrix, and $a$ is single-layer feed-forward neural network. Then the attention weight $\alpha_{ij}$ for nodes $j \in \mathcal{N}_i$ is computed as

$$\alpha_{ij} = \frac{\exp(e_{ij})}{\sum_{k \in \mathcal{N}_i} \exp(e_{ik})}, \quad (3)$$

where $\mathcal{N}_i$ is the neighborhood of node $i$ in the graph. At last, node $i$'s feature $h_i'$ is updated as

$$h_i' = f\left(\sum_{j \in \mathcal{N}_i} \alpha_{ij} \mathbf{W} h_j\right), \quad (4)$$

where $f$ is the SoftPlus function which is a smooth approximation to the ReLU function. Multi-head attention[57] is also used where $K$ independent attention mechanisms are executed and their feature vectors are averaged as

$$h_i' = f\left(\frac{1}{K} \sum_{k=1}^{K} \sum_{j \in \mathcal{N}_i} \alpha_{ij}^k \mathbf{W}^k h_j\right). \quad (5)$$

The final feature embedding vectors of the structure feature encoders are obtained by applying global mean pooling operation on node features.

In this work, we use an MLP as the label encoder.

*Probabilistic feature and label embeddings.* In this work, both features and labels are embedded into a latent Gaussian mixture space and the feature and label embeddings are aligned to exploit feature and label correlations. Therefore, latent variables $Z_F$ and $Z_L$ are assumed to follow some mixture of multivariate Gaussian distributions:

$$Z_F \sim \sum_{i=1}^{K} \pi_i \mathcal{N}_i(Z_F | \mu_i, \text{diag}(\sigma_i^2)) \text{ and}$$
$$Z_L \sim \sum_{j=1}^{K'} \pi_j' \mathcal{N}_j'(Z_L | \mu_j', \text{diag}(\sigma_j'^2)), \quad (6)$$

where $K$ and $K'$ are the presumed number of clusters in the latent space and $\pi_i$ and $\pi_j'$ are the learned prior probabilities of related clusters. In this work, $K$ is 10 and $K'$ equals the dimension of label vectors, and the dimensionality of the Gaussians is set to be 128. Since both $Z_F$ and $Z_L$ are assumed to follow a mixture of multivariate Gaussian distributions, we use KL divergence to align them, which is not analytically tractable. However, we can optimize the KL divergence between two Gaussian mixtures by optimizing the following upper bound $\mathcal{L}_{KL}$[58]:

$$\text{KL}(Z_F || Z_L) \leq \mathcal{L}_{KL} = \sum_{i,j} \pi_i \pi_j' \text{KL}(\mathcal{N}_i, \mathcal{N}_j'). \quad (7)$$

*Supervised contrastive learning.* A shared translator $f_d(\cdot)$ extracts representation vectors $H_F$ and $H_L$ from embeddings $Z_F$ and $Z_L$, respectively. In order to further facilitate learning a label-aware feature representation $H_F$, we adopt supervised contrastive learning:

(i) Maximizing agreement between feature and label representations: a projector, denoted as $f_h$, transforms the feature and label representations and matches them with each other using a contrastive loss, where the projector is an MLP with one hidden layer. Concretely, let $B_F = f_h(H_F)$ and $B_L = f_h(H_L)$.

We define a batch of samples' feature and label representation pairs as $\mathcal{B} = \{(B_X, B_Y)\}$. We then train the decoder and projector and use the following contrastive loss to maximize agreement between feature and label representations:

$$\mathcal{L}_D = -\frac{1}{|\mathcal{B}|} \sum_{(B_X, B_Y) \in \mathcal{B}} \log \frac{\exp(B_X^\top B_Y / \tau)}{\sum_{(B_X', B_Y') \in \mathcal{B}'} \exp(B_X^\top B_Y' / \tau)} \quad (8)$$

where $\mathcal{B}' = \{\mathcal{B} \setminus \{(B_X, B_Y)\}\}$ and $\tau \geq 0$ is the temperature. Note that it is empirically beneficial to define the contrastive loss on the projections $B_X$ and $B_Y$ rather than the representations $H_X$ and $H_Y$[38].

(ii) Label prediction: A predictor, denoted as $f_p$, transforms the feature and label representations and matches them to the label $L$. Let $A_F = f_p(Z_F)$ and $A_L = f_p(Z_L)$. We define a symmetrical loss to learn a multi-property regressor:

$$\mathcal{L}_C = \frac{1}{2} \text{LOSS}(A_F, L) + \frac{1}{2} \text{LOSS}(A_L, L), \quad (9)$$

**Table 3 Mat2Spec Pseudocode.**

| | | |
|---|---|---|
| **Input**: | $\{(X_i, Y_i)\}_{i=1}^N$, $\lambda_1$, $\lambda_2$, and $\lambda_3$. | |
| **Output**: | Feature encoder $f_e(\cdot)$, decoder $f_d(\cdot)$, and predictor $f_p(\cdot)$. | |
| 1: | **for** $(X, Y)$ in dataloader **do** | |
| 2: | $Z_F, Z_L = f_e(X), f_l(Y)$. | ▷latent codes |
| 3: | $H_F, H_L = f_d(Z_F), f_d(Z_L)$. | ▷ representations |
| 4: | $A_F, A_L = f_p(H_F), f_p(H_L)$. | ▷ predictions |
| 5: | $B_F, B_L = f_h(H_F), f_h(H_L)$. | ▷projections |
| 6: | Compute the contrastive loss $\mathcal{L}_D$ according to equation (8). | |
| 7: | Compute the symmetrical loss $\mathcal{L}_C$ according to equation (9). | |
| 8: | Compute the KL loss $\mathcal{L}_{KL}$ according to equation (7). | |
| 9: | $\mathcal{L} = \lambda_1 \mathcal{L}_D + \lambda_2 \mathcal{L}_C + \lambda_3 \mathcal{L}_{KL}$. | ▷ combine losses |
| 10: | $\mathcal{L}$. backward (). | ▷ compute gradient |
| 11: | update$f_e(\cdot), f_l(\cdot), f_d(\cdot), f_p(\cdot)$, and $f_h(\cdot)$. | ▷ update parameters |
| 12: | **Return**$f_e(\cdot)$, $f_d(\cdot)$, and $f_p(\cdot)$. | |

where LOSS( · ) denotes the loss function for the given setting: KL, MSE, MAE, or WD.

The final loss function used in Mat2Spec is a combination of $\mathcal{L}_D$, $\mathcal{L}_C$, and $\mathcal{L}_{KL}$, which is defined as follows:

$$\mathcal{L} = \lambda_1 \mathcal{L}_D + \lambda_2 \mathcal{L}_C + \lambda_3 \mathcal{L}_{KL}, \qquad (10)$$

where $\lambda_1 = 1$, $\lambda_2 = 0.1$, and $\lambda_3 = 1.1$ control the weights of the three loss terms.

*Implementation and hyperparameters.* The crystal feature encoder has two four-head attention layers. Each layer of the crystal feature encoder has 103 nodes where each node corresponds to an element and is represented by a 128-dimensional vector. The label encoder for learning mixing coefficients is a two-layer MLP, and the two layers have 128 and $K'$ neurons, respectively, where $K'$ is the length of the label vector ($K' = 51$ for the phDOS and $K' = 128$ for the eDOS). The translator is a two-layer MLP, and the 2 layers have 512 and 512 neurons, respectively. The projector is a two-layer MLP, and the two layers have 512 and 1024 neurons, respectively. The predictor is a two-layer MLP, and the two layers have 256 and $K'$ neurons, respectively. The number of multivariate Gaussians $K$ produced by the feature encoder is 10 and the dimension $D$ of each multivariate Gaussian is 128. The dimensions of representation and projection vectors are 128 and 1024, respectively.

We used grid search for hyperparameter optimization, where we set the batch size and the number of training epochs to 128 and 200, respectively, for all experiments. The learning rate was chosen from 0.0005 to 0.01 in steps of 0.0005, dropout ratio from [0.3, 0.5, 0.7], and weight decay ratio from [0, 0.01, 0.001, 0.0001]. In this work, we use the same set of hyperparameters for all experiments where learning rate, dropout ratio, and weight decay ratio are set to be 0.001, 0.5, and 0.01, respectively. Our model was implemented with the Pytorch deep learning framework and the whole model was trained with the AdamW optimizer using a 0.01 weight decay ratio in an end-to-end fashion in a machine with NVIDIA RTX 2080 10GB GPUs.

**Data generation.** The phDOS dataset was replicated from ref. [14] to facilitate direct comparison between the models. The dataset is randomly divided into 1220, 152, and 152 samples for training, validation, and test sets, respectively. The original phDOS is presented in ref. [34] publicly available through the MP website. We remind here that the phDOS from this dataset were cut at a frequency of 1000 cm$^{-1}$, smoothed via a Savitzky–Golay filter, and interpolated on a common 51-frequencies grid. The calculation of $C_V$ and $\bar{\omega}$ similarly proceeded in the same way as this prior work by using the correspondent functions present in pymatgen[59].

The eDOS dataset was acquired from the Materials Project (version 2021-03-22), where the eDOS values are computed according to a DFT recipe reported in the MP documentation[60]. The dataset contains eDOS and crystal structures of non-magnetic materials, both metallic and semiconductors/insulators, with an energy grid of 2001 points (VASP NEDOS set to 2001). This dataset has been randomly divided into training, validation, and test sets, containing 80,10,10% of the samples, respectively. Another dataset that contains only the structures of materials with no available eDOS in the MP was acquired for prediction. Only materials which are flagged as non-magnetic by MP were considered. These two datasets contain about 30,000 and 24,000 entries, respectively.

The energy range of the eDOS can be very large and varies among the MP entries. To consider a consistent energy grid the eDOS from MP was resampled. The energy range taken into account is the first 4 eV below the valence bands maximum and above the CBM, a range chosen based on common uses of eDOS for studying related material properties. Also, for the reasons mentioned above, we removed the band gap from the eDOS, when present, making the conduction band start just after the valence band. Upon this cutting step, the energy range was divided into 128 bins and the average of the eDOS values in each bin was taken.

**Calculation of eDOS.** The eDOS was calculated for materials with no available eDOS in the MP that were predicted to have a band gap in the valence band. The 32 Materials Project entries are as follows: mp-1236246, mp-1236485, mp-1222301, mp-1227246, mp-1227245, mp-1217599, mp-1187874, mp-1222008, mp-1185617, mp-1206725, mp-1184817, mp-1185314, mp-1220591, mp-1184222, mp-1180477, mp-1147636, mp-1094477, mp-1519062, mp-1518752, mp-1518664, mp-1222270, mp-1185435, mp-1185407, mp-1185397, mp-1111899, mp-1111898, mp-1111559, mp-1094298, mp-1039010, mp-1522162, mp-1519126, mp-1520737. The DFT computations were performed following the MP recipe by means of the atomate package[61] so that the eDOS are computed in the same way as those used for model training.

The criteria outlined in the eDOS VB gap use case were applied to the computed DFT eDOS to verify the predictions. We note that the same criteria with different parameters can be applied to find gaps in the conduction band.

## Data availability

The input data as well as the predicted phDOS and eDOS data generated in this study have been deposited in the CaltechData database under accession code 8975 and https://doi.org/10.22002/D1.8975, and are available at https://data.caltech.edu/records/8975 and https://www.cs.cornell.edu/gomes/udiscoverit/?tag=materials.

## Code availability

Source code for Mat2Spec[62] is available from https://github.com/gomes-lab/Mat2Spec(https://doi.org/10.5281/zenodo.5863471) and from https://www.cs.cornell.edu/gomes/udiscoverit/?tag=materials.

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

## Acknowledgements

This work was funded by the U.S. Department of Energy, Office of Science, Office of Basic Energy Sciences, under Award DE-SC0020383 (design of prediction task, development of use case, validation of predicted materials, model evaluation; grant received by J.N., C.G., and J.G.) and by the Toyota Research Institute through the Accelerated Materials Design and Discovery program (development of machine learning models; grant received by C.G. and J.G.).

## Author contributions

S.K. designed and implemented Mat2Spec with guidance from C.G., F.R., D.G., J.N., and J.G. designed the use case. F.R. performed DFT calculations and interpreted results. S.K., F.R., D.G., and J.G. wrote the manuscript with contributions from all authors. J.N., C.G., and J.G. conceived the project and supervised the work.

## Competing interests

The authors declare no competing interests.
