## [Peer Review File · Nature Communications]

REVIEWER COMMENTS

Reviewer #1 (Remarks to the Author):

The authors present a new supervised machine learning (ML) model for spectral properties, named Mat2Spec. It is based on a series of existing state of the art techniques, including graph attention networks, probabilistic embeddings and contrastive learning. The model is uncommon as it focuses on material's spectral properties, such as the phonon density of states and electronic density of states, instead of single scalar properties. The model significantly outperforms current models on the phonon DOS and eDOS; but given the complexity of the problem, the observed accuracies are still far from perfect. An important use case on thermoelectrics and transparent conductors is given, with a selection of hypothetical candidates.

The paper is well written, follows common ML evaluation standards, and contains reproducible results with the provided code.

Given that the main novelty is the model architecture, it might preferentially be considered in a more specialised journal as npj. Comput. Mater. In either case, the following comments need to be addressed before publication:

Concerning the methodology - which is the main novelty of the paper, some details are still unclear to me:

- Why are "two" successive embeddings used: (i) a probabilistic (Z_f and Z_l) embedding and (ii) representation vectors H_f and H_l ? Contrastive learning models typically have a single embedding (e.g. SimCLR). Please explain briefly.
- For the probabilistic embedding (line 322), what values are chosen for μ and σ ?
- The dimensions of the different vectors between the sub-models are unclear to me. For instance, the label encoder seem to have an output vector of size 128 (line 333), but the number of learnable parameters for the multivariate Gaussian sum is set to K' , with equals the label dimension (line 322), which seem to be of length 51 (line 347). Please clarify by for instance adding input/output dimensions in "figure 1: architecture ...", or rename figure 1 as "model schematic ..." and add a separate architecture figure.
- Between line 327 and 328 (line number is missing): The label prediction is defined as a function f_p , that takes the probabilistic embedding Z_f and Z_l as input. Figure 1 suggest that only the representation vector H_f is used. Please correct the text or figure.

Concerning the results:

- Line 132: use “scaled” instead of “normalized”
- Line 148: please provide the temperature of Cv
- Line 246 and corresponding paragraph / Table 2: DFT computations were made on the 32 predicted positive samples. Therefore the recall has no sense here, as all positives are unknown from the considered set. Use the precision instead. Similarly, Table 2 is quite confusing as it is computed on the “biased” 32 predicted positive subset instead of the screened subset. It might be necessary to indicate as a footnote that although FN equals zero, some positive samples might be missing from the screened set.
- Line 256: typo on “the”.
- Line 294: Reference doesn't correspond.

Reviewer #2 (Remarks to the Author):

The manuscript by Kong et al. presents a graph-neural-network based approach for the prediction of phonon and electronic density of states for crystalline materials, given their composition and atomic structure.

The methodology is rather involved, being the architecture of the network composed of an encoder module (a probabilistic embedding generator) and a decoder module (based on contrastive learning), all of which is state-of-the-art technology in the neural network field. The key challenge of the presented work is in predicting spectral properties of materials, i.e., arrays (specifically, histograms) rather than the more usual scalar properties (e.g., formation energy, band gaps).

The paper is well written and clearly the presented approach outperforms everything else on the market, on the dataset selected by the authors, which, despite the limitation of the the specific choice, is rather broad and accessible for future challenges.

As much as this work deserves publication, if nothing else to spark further improvement in the field, I do not see

a sufficient novelty in the methodology or a deep insight in the results and therefore do not recommend this manuscript for publication in Nature Communications.

Here are my major concerns:

- the less good prediction, in particular in eDOS, and I am referring to quartiles 5 and 4 in Fig. 3 miss important structure in particular in the valence states. I would try to understand the reason of such

discrepancy, e.g., if it is related in some sense to the distance between the test points in these higher quartiles and the training points or if there are eDOS shapes that are intrinsically more difficult to learn? If the latter, why?

- it is even unclear from a physical point of view what is the merit of reproducing conduction-band electronic states of a DFT calculation. Are they supposed to map reliably into any observable physical quantities?

- the performance of the classification task (VB gap discovery) seems pretty poor. Half of the predictions are wrong and the way negatives are counted is weird. In principle, one should have gone through the (100-2.8)% materials that are not selected as candidates (i.e., the negatively predicted) and count how many are true and how many are false. Similarly, testing the other methods only on the materials that are selected by Mat2Spec to conclude that they introduce only (true and false) negatives and not a single positive is misleading. They could have predicted other true positives over all the tested materials.

- the input of the model requires atomic positions. So, the new, test materials required to be already known at least in terms of geometrical structure. As much as calculating a eDOS is expensive (at least, more expensive than running an NN), such calculation is implicitly done when the geometrical structure has been optimized in the first place. Could one use an approximate geometrical structure as input and still learn the optimized eDOS? This could be useful for materials screening.

- to be extremely clear and honest: outperforming existing (bad) models is not enough. Physics is not about being having a model that is better more often than existing ones, it is rather about getting a model that makes reliable predictions over a well defined domain of applicability. I.e, it is acceptable to have a model that does not cover some cases, but one should know by means of some descriptors which cases are not well (or not at all) described. Plugging everything into a machine-learned model, and the observing a distribution of errors, some of which too large to be of any predictive use is highly not satisfactory. Regardless of how elaborate the learning algorithm and resulting predictive model is.

- a minor comment about the methodology: the activation function (eq. 4) should be specified, not just hinted at: "such as the sigmoid". It has been shown that it matters what activation function is used. E.g., the most commonly used these in the recent years is ReLU, but more recently people started tailoring the activation function depending on the task.

Reviewer #3 (Remarks to the Author):

This paper describes a machine learning model with an architecture tailored to address an important challenge in computational materials science: the prediction of spectra. The proposed approach, named Mat2Spec, predicts spectra (e.g., phonon, electronic densities of state) from material's

structure by exploiting the fact that points along these spectra are highly correlated and using a new approach that employs “contrastive learning” to provide further information when training the weights. The work is highly innovative, and the authors take exceptional care in placing the work in context with the community and illustrating its application to materials research. I recommend it to be published with only minor revisions.

One way I think the paper can be improved is by providing evidence to illustrate the importance of the contrastive learning. The paper establishes the importance of the choice of normalization and training loss function but does not evaluate the effect of other design choices in the architecture. Considering the complexity the contrastive losses add to the approach and, of a lesser importance, the prominence of “contrastive learning” in the title, I would advocate for running some experiments without the contrastive loss or the “alignment” loss of the embedding Gaussians.

The other recommendation I have for the paper is to clarify a few points of the architecture. These include:

- What are the dimensionality of the Gaussians? Are they equal to the number of points in the spectrum output?
- Correspondingly, does this mean the output embeddings (e.g., Z_F) have a shape of (batch size, number of output points, number of output points)? If so, I can better see how the “translator” and “predictor” layers exploit the correlation between rows in the innermost dimension of the tensor imposed by the Gaussians. If that is a correct line of reasoning, could you describe this logic to the reader? (If not, could you better describe the link between how covariances in the Gaussian propagate to correlations in the predicted labels?)
- Are the embeddings generated by sampling from the Gaussians?
- Describing in the text how the model is used for inference (i.e., only the embedding produced from the GNN encoding of the material are needed) would be helpful to reinforce the point to the reader.
- Explaining the concepts behind the Wasserstein and KL losses would be helpful to non-expert readers to understand why they are better choices than a conventional MSE loss.

Beyond these main comments, I have a few other points of feedback:

- Briefly introducing the datasets and their usefulness before the Phonon DOS and Electronic DOS sections would improve the reader’s ability to quickly understand the results.
- The use of derived metrics, such as the heat capacity or classification performance, to quantify model performance really help the reader understand the quality of the models. The use of representative cases from the different error quantiles is also a clever approach to showing the model performance in a less-biased way.

REVIEWER COMMENTS

Blue text: Responses
Olive text: Manuscript revisions

Reviewer #1 (Remarks to the Author):

The authors present a new supervised machine learning (ML) model for spectral properties, named Mat2Spec. It is based on a series of existing state of the art techniques, including graph attention networks, probabilistic embeddings and contrastive learning. The model is uncommon as it focuses on material's spectral properties, such as the phonon density of states and electronic density of states, instead of single scalar properties. The model significantly outperforms current models on the phonon DOS and eDOS; but given the complexity of the problem, the observed accuracies are still far from perfect. An important use case on thermoelectrics and transparent conductors is given, with a selection of hypothetical candidates.

The paper is well written, follows common ML evaluation standards, and contains reproducible results with the provided code.

Given that the main novelty is the model architecture, it might preferentially be considered in a more specialised journal as npj. Comput. Mater. In either case, the following comments need to be addressed before publication:

Response: Thank you for recognizing the novelty of our work, importance of our use case, and excellent performance of Mat2Spec compared to prior art. Our dedication to publishing our work in a journal with broad readership arises from the generality of our approach. The efficient encoding of knowledge from spectral data with the Mat2Spec architecture has broad applicability in materials science, physics, and chemistry, especially given our demonstration that the model architecture enables deployment with much smaller training sets than prior art. More generally, data in the form of probability distributions can be found in all branches of science and technology. Many deep learning techniques overfit the training data in such settings, which limits the generalization of the models and thus the ability to make predictions far from the training data. We believe that this work will inspire researchers to develop more effective machine learning techniques for predicting spectra and probability distributions, and that Mat2Spec can be generalized to other domains of multi-output regression. Specifically, our probabilistic embedding generators allow us to have significant control over how we want to model our latent distribution via a prior distribution. Therefore, our model can be generalized to other domains by choosing appropriated prior distributions, model alignment losses, and prediction losses. For example, we can choose Poisson loss for counting problems and binary cross entropy loss for classification problems. We have expanded this discussion of the generality and extension to new problems in the Discussion.

Concerning the methodology - which is the main novelty of the paper, some details are still unclear to me:

- Why are “two” successive embeddings used: (i) a probabilistic (Z_f and Z_l) embedding and (ii) representation vectors H_f and H_l ? Contrastive learning models typically have a single embedding (e.g. SimCLR). Please explain briefly.

Response: This is an excellent question about the construction of latent representations, which is designed based on our machine learning strategies. Note that, for example, Z_f and Z_l are used in training but only Z_f is used during prediction. The embeddings Z_f and Z_l are used to exploit hidden label structures [1] and the second representations H_f and H_l are further used to maximize the agreement between the feature and label representations [2]. We further give the following specifications:

- (1) The first feature and label probabilistic embeddings (Z_f and Z_l) are designed to exploit feature and label correlations implicitly. This is inspired by the label embedding technique [1] where it encodes both instances and their labels onto a structured latent space to align the instance embedding with the corresponding label embedding. While the traditional label embedding techniques assume a deterministic latent space which lacks feature and label representation smoothness and thus could be sensitive to noise, Mat2Spec learns a probabilistic latent space where we can have significant control over how we want to model our latent distribution via a prior distribution.
- (2) The second representation vectors (H_f and H_l) are designed to maximize the agreement between the two representations, which are typical components of contrastive learning [2]. Different from traditional contrastive learning methods that use image augmentations to obtain different representations, H_f and H_l are extracted by a shared translator from Z_f and Z_l in our model, respectively.

The discussion of these topics has been expanded in the Methods.

[1] K.-H. Huang and H.-T. Lin. Cost-sensitive label embedding for multi-label classification. *Machine Learning*, 106(9):1725–1746, 2017.

[2] Chen, Ting, et al. "A simple framework for contrastive learning of visual representations." *International conference on machine learning*. PMLR, 2020.

- For the probabilistic embedding (line 322), what values are chosen for μ and σ ?

Response: Implementing the probabilistic embeddings described in the prior response involves learning μ and σ from data during the end-to-end model training.

- The dimensions of the different vectors between the sub-models are unclear to me. For instance, the label encoder seem to have an output vector of size 128 (line 333), but the number of learnable parameters for the multivariate Gaussian sum is set to K' , with equals the label dimension (line 322), witch seem to be of length 51 (line 347). Please clarify by for instance

adding input/output dimensions in “figure 1: architecture ...”, or rename figure 1 as “model schematic ...” and add a separate architecture figure.

Response: Thanks for pointing out the need to improve clarity and for the excellent suggestion to include this information in the model architecture figure. We have updated the description of the label encoder, translator, and projector and additionally noted all dimensions in Figure 1. The label encoder has an output vector (mixture coefficients) of size K' (i.e., $K'=51$ for the pHDOS and $K'=128$ for the eDOS). The dimensions of the multivariate Gaussians are all 128. The dimensions of embedding vectors Z_f and Z_l are both 128 as well. The dimensions of feature and label representation vectors H_f and H_l are both 512. The dimension of feature and label projection vectors B_f and B_l are both 1024.

- Between line 327 and 328 (line number is missing): The label prediction is defined as a function f_p , that takes the probabilistic embedding Z_f and Z_l as input. Figure 1 suggest that only the representation vector H_f is used. Please correct the text or figure.

Response: Yes, you are right and we have updated Figure 1.

Concerning the results:

- Line 132: use “scaled” instead of “normalized”

Response: Fixed.

- Line 148: please provide the temperature of C_v

Response: Fixed.

- Line 246 and corresponding paragraph / Table 2: DFT computations were made on the 32 predicted positive samples. Therefore the recall has no sense here, as all positives are unknown from the considered set. Use the precision instead. Similarly, Table 2 is quite confusing as it is computed on the “biased” 32 predicted positive subset instead of the screened subset. It might be necessary to indicate as a footnote that although FN equals zero, some positive samples might be missing from the screened set.

Response: This is an insightful observation that motivated a more balanced investigation of discovering VB gap materials in the set of mp-ids with no available eDOS. By nature of ML-guided discovery in this space for which DFT calculation of every material is untenable, the selection of materials for DFT validation must be “biased”, but we alleviated this bias by calculating eDOS via DFT for 49 random mp-ids, which were all predicted as negatives by Mat2Spec. To make a better comparison with GATGNN-KL (the best performing model based on prior art) we have expanded the set of DFT calculations to include all of its predicted positives as well. The full list of mp-ids and classification results are included as a csv file in the SI. The updated study further highlights the excellent absolute and relative performance of Mat2Spec, as indicated by the following new paragraph and revised table:

Querying the Materials Project for materials that lack eDOS and meet the energy above hull, number of elements, and band gap requirements produced 8,106 candidate materials. DFT calculations were performed on 100 of these materials, including 49 randomly-selected materials as well as the 51 materials predicted to have a VB gap by either the Mat2Spec SumNorm-KL or GATGNN SumNorm-KL models. Table 2 shows the excellent performance of Mat2Spec with precision and recall of 0.47 and 1.0, respectively, both exceeding the GATGNN values of 0.13 and 0.2, respectively. There are 4 materials for which both Mat2Spec and GATGNN predict a VB gap, with 3 of these validated as TP by DFT. There are 19 materials predicted positive by GATGNN but negative by Mat2Spec, and none of these were found to have a VB gap via DFT, i.e. Mat2Spec correctly predicted all of these as negatives. Of the 28 materials predicted positive by Mat2Spec but negative by GATGNN, 12 were found to have a VB gap via DFT, i.e. only Mat2Spec enabled discovery of these 12 gapped metals. Of the randomly selected materials, none were found to have a VB gap, which is commensurate with the expectation that the positivity rate in the set of candidate materials is similarly low as that of the test set. Collectively the results show the excellent performance of Mat2Spec for the discovery of these rare materials, where 15 discoveries were validated by DFT from Mat2Spec's predicted positives. This down-selection by a factor of 253 from the set of candidates corresponds to a proportional savings in the computation time for discovering gapped metals.

ML model	Setting		eDOS prediction				VB gap identification			VB gap discovery	
	Scaling	Loss	R ²	MAE	MSE	WD	F1	precision	recall	precision	recall
E3NN	Standard	MAE	0.39	5.24	105.1	0.48	0.035	0.333	0.019	-	0
GATGNN	Standard	MAE	0.30	4.89	120.9	0.42	0.182	0.263	0.139	0	0
Mat2Spec	Standard	MAE	0.53	3.64	80.4	0.27	0.352	0.509	0.269	0.67	0.27
E3NN	SumNorm	WD	-2.1	9.81	542.8	0.40	0	0	0	-	0
GATGNN	SumNorm	WD	0.19	6.41	140.1	0.26	0.018	0.200	0.009	0	0
Mat2Spec	SumNorm	WD	0.38	5.23	107.7	0.29	0.018	0.250	0.009	-	0
E3NN	SumNorm	KL	0.41	5.01	101.9	0.47	0	0	0	-	0
GATGNN	SumNorm	KL	0.32	4.89	118.2	0.35	0.243	0.450	0.167	0.13	0.20
Mat2Spec	SumNorm	KL	0.57	3.8	74.5	0.21	0.397	0.698	0.278	0.47	1.00

Table 2: Results of eDOS prediction. For each combination of 3 ML models and 3 settings, the performance metrics include 4 measures of the prediction of the full 128-D eDOS (evaluated on the test set), 3 measures of the classification accuracy for identification of gaps in the eDOS in the VB (evaluated on the test set), and 2 measures of the classification accuracy for 100 materials with no prior available eDOS. ... The 100 materials for which we performed DFT calculations include 49 random selections as well as predicted positives from GATGNN and Mat2Spec in the SumNorm-KL setting. Precision values are missing for 4 models with no positive predictions among the 100 materials.

- Line 256: typo on "the".

Response: Fixed.

- Line 294: Reference doesn't correspond.

Response: Fixed.

Reviewer #2 (Remarks to the Author):

The manuscript by Kong et al. presents a graph-neural-network based approach for the prediction of phonon and electronic density of states for crystalline materials, given their composition and atomic structure.

The methodology is rather involved, being the architecture of the network composed of an encoder module (a probabilistic embedding generator) and a decoder module (based on contrastive learning), all of which is state-of-the-art technology in the neural network field. The key challenge of the presented work is in predicting spectral properties of materials, i.e., arrays (specifically, histograms) rather than the more usual scalar properties (e.g., formation energy, band gaps).

The paper is well written and clearly the presented approach outperforms everything else on the market, on the dataset selected by the authors, which, despite the limitation of the the specific choice, is rather broad and accessible for future challenges.

As much as this work deserves publication, if nothing else to spark further improvement in the field, I do not see

a sufficient novelty in the methodology or a deep insight in the results and therefore do not recommend this manuscript for publication in *Nature Communications*.

Response: We appreciate the overall positive assessment of the paper as well as the series of insightful comments that guided much of our revisions. One important outcome is the demonstrated ability of Mat2Spec to transfer knowledge from the full breadth of materials to improve predictions for a specific subclass of materials. We believe that the totality of our revisions, described below, demonstrates even greater utility and novelty of Mat2Spec, and makes an even more compelling case for publication in *Nature Communications*.

Here are my major concerns:

- the less good prediction, in particular in eDOS, and I am referring to quartiles 5 and 4 in Fig. 3 miss important structure in particular in the valence states. I would try to understand the reason of such discrepancy, e.g., if it is related in some sense to the distance between the test points in these higher quartiles and the training points or if there are eDOS shapes that are intrinsically more difficult to learn? If the latter, why?

Response: This is an important topic that touches upon interpretability, diagnostics, and other frontier challenges in deep learning. We tackled this question on 4 fronts, studying the distribution of loss as a function of descriptors for (i) material complexity, (ii) similarity to training data, (iii) component frequencies in the eDOS signal, and (iv) specific materials subclasses. The results are largely intuitive and point to a prevalence of higher prediction loss for a certain range of each descriptor, which can be used to assess the trustworthiness of predictions when

deploying Mat2Spec, e.g. for predicting eDOS of new materials. The variability in performance with different materials subclasses is discussed further below, where we additionally show that specific eDOS features that are not well-modelled by Mat2Spec (related to your comment “miss important structure”) are better modelled by fine-tuning the Mat2Spec model for a subclass of materials. The new text in the manuscript and new figures in the SI are as follows:

In the Supporting Information, we explore the underpinnings of the distribution in prediction accuracy of Mat2Spec in the SumNorm-KL setting. Intuitively, the prediction task is harder and the MAE is generally larger when (i) the material's structure is more complicated, (ii) there are few examples of similar materials in the training set, and/or (iii) the eDOS contains high frequency features, which are all demonstrated in Figures S3 and S4. Figure S5 illustrates that the distribution of prediction quality can vary substantially with materials class, which is likely due to a convolution of the aforementioned effects with the chemical complexity of a given subclass of materials.

Figure S3. Mat2Spec loss vs. material complexity: Each panel contains a point for each test set material using the Mat2Spec SumNorm-KL setting. The prediction loss is plotted as a function of 2 descriptors for the complexity of a material, (top) number of elements and (bottom) number of sites in the unit cell for each material. The expected positive correlation is observed between material complexity and prediction loss. For either the number of elements or the number of sites, the available training data with a similar number of elements or sites also decreases as these numbers increase. Having few training examples of materials with similar complexity exacerbates the challenge of predicting the properties of complex materials. These results inform the trustworthiness of predictions for new materials based on these simple descriptors.

Figure S4. Mat2Spec loss vs. similarity of training data: Each panel contains a point for each test set material using the Mat2Spec SumNorm-KL setting. The prediction loss is plotted as a function of 2 descriptors for the similarity to materials in the training data, the number of materials in the training set that have (top) the same set of elements and (bottom) the same spacegroup for each material in the test set. The expected negative correlation is observed between the amount of similar training data and the prediction loss. The relationship is more profound for the number of training examples with the same elements, suggesting that training examples with the same elements in different compositions and/or structures facilitates the model's learning of the chemistry of those elements. These results inform the trustworthiness of predictions for new materials based on these simple descriptors.

Figure S5. Variation in loss among subclasses: Since many materials discovery projects are focused on a particular subclass of materials, the variation in performance of Mat2Spec among different subclasses is shown. The 3 example subclasses are (top) materials containing 3 non-rare-earth elements, (middle) the subset of those that contain oxygen, and (bottom) materials with spacegroup 225. For each subclass, the distribution of losses for test set materials is shown with respect to the MAE and WD quintiles of Mat2Spec SumNorm-KL predictions (the quintiles of the full test set, whose ranges are noted on each horizontal axis). A subclass with the same distribution of losses as the full test set will have equal counts in each quintile. The first subclass has a fairly uniform distribution of loss, while the subset of this subclass for oxygen-containing materials has a distribution substantially skewed toward higher-loss quintiles. Commensurate with the observation from Figure S4, where having many training examples with the same spacegroup leads to lower prediction loss, the distribution of loss for the subclass of materials with the most populous spacegroup (spacegroup 225) is strongly skewed toward the lower quintiles. Further study is required to elucidate whether the descriptors in Figures S3 and S5 explain the distribution of loss within a given subclass, or whether it is the complexity of the materials chemistry with the subclass that underlies the difficulty of the prediction task. The latter appears to be true for ternary oxides.

Figure S6. Mat2Spec loss vs. Fourier components: The primary observation from inspection of the quintile plots of MAE loss for Mat2Spec SumNorm-KL is that sharp peaks and other high-frequency features are sometimes poorly predicted. To study the generality of this observation, the Fourier transform of each test set eDOS was calculated and divided into 5 frequency ranges. The relative intensity in each of these frequency ranges characterizes whether a given material has primarily low, high, etc. features. The plotted loss in each panel is the (same) prediction loss for the test set materials, and the Pearson correlation coefficient (r) is shown in each panel. For both loss metrics, a moderate negative correlation is observed for the lowest frequency range, and moderate positive correlation is observed for each of the higher frequency ranges. The high level observation is that while Mat2Spec can accurately predict eDOS with high frequency features for some materials, the existence of high frequency features does predispose materials to having higher prediction loss.

- it is even unclear from a physical point of view what is the merit of reproducing conduction-band electronic states of a DFT calculation. Are they supposed to map reliably into any observable physical quantities?

Response:

The conduction bands computed with DFT represent a valuable approximation to the true electronic structure of unoccupied states, observables that can be directly measured, for example, by inverse photoemission spectroscopy. In addition, the unoccupied spectrum of a material underpins its optical and transport properties, which are critical to the functionality of optoelectronic materials. DFT calculations of conduction band eDOS therefore guide the identification of interesting new materials and serve as a stepping stone for more accurate (and expensive) electronic structure calculations. Despite the limitations of standard approximate exchange-correlation functionals, there is substantial literature precedent for using DFT band structures (including conduction bands) to understand and predict electronic properties of materials. In our manuscript, the eDOS training data is from DFT, and Mat2Spec inherits the associated limitations; while, as stated above, the eDOS at this level of theory is still very valuable, we would also note that the success of Mat2Spec in this case is proof of principle that it could be applied to a database of even more accurate spectra, should it be available.

- the performance of the classification task (VB gap discovery) seems pretty poor. Half of the predictions are wrong and the way negatives are counted is weird. In principle, one should have gone through the (100-2.8)% materials that are not selected as candidates (i.e., the negatively predicted) and count how many are true and how many are false. Similarly, testing the other methods only on the materials that are selected by Mat2Spec to conclude that they introduce only (true and false) negatives and not a single positive is misleading. They could have predicted other true positives over all the tested materials.

Response: We agree that our DFT validation of only Mat2Spec's predicted positives provided an inadequate assessment of the absolute and relative performance of Mat2Spec for discovering materials with a VB gap. We expanded the set of DFT validations to 100 mp-ids including GATGNN's predicted positives as well as random selections. Please see the response to Reviewer #1's similar comment where we provide the new text and updated Table 2. The expanded results further highlight the excellent performance of Mat2Spec.

- the input of the model requires atomic positions. So, the new, test materials required to be already known at least in terms of geometrical structure. As much as calculating a eDOS is expensive (at least, more expensive than running an NN), such calculation is implicitly done when the geometrical structure has been optimized in the first place. Could one use an approximate geometrical structure as input and still learn the optimized eDOS? This could be useful for materials screening.

Response:

This is an excellent topic that touches upon the use cases for Mat2Spec.

We note that while DFT-based structure optimization can be a demanding computational task especially when the number of atoms in the unit cell is large, a high-quality eDOS does not automatically follow from the relaxation and requires additional computational expense. Specifically, computing an accurate eDOS requires two steps, (i) a self-consistent calculation of the ground-state charge density with a relatively sparse k-point mesh followed by (ii) a non-self-consistent calculation of eigenvalues on a dense k-point mesh. The computational expense of the first step depends on the initial (unrelaxed) structure. The second step is linear in the number of k-points and can be quite expensive, beyond the relaxation, if a large number of k-points is required.

We note that important aspects of our work include our determination of the size of the training dataset required to create a suitable surrogate model for eDOS. We have demonstrated a model architecture that is designed to operate in small-data regimes, where it dramatically outperforms state of the art. Having shown here that 2,000-3,000 training examples can yield an eDOS prediction model that generalizes well, we can, in a follow-on study, generate higher quality eDOS training data, which will lead to greater computational savings when using Mat2Spec. We have highlighted this opportunity with the following addition:

The ability to make meaningful predictions with data sizes on the order of 10^3 materials is critical for predicting spectral properties that are much more expensive to acquire, where the increased expense typically corresponds to less available training data than the eDOS dataset used in the present work. The increased expense also expands the opportunity for accelerating materials discovery with Mat2Spec.

That being said, the reviewer makes an intriguing point regarding the prediction of eDOS for materials that have not undergone DFT structure relaxation. In our revised manuscript, we address this question in part by considering the “unrelaxed” version of the structures in the test set; that is, we predict the eDOS using the initial CIF file before DFT structure relaxation for each mp-id (this is the only type of unrelaxed CIF file that the Materials Project has for each mp-id). We assess the degree to which our prediction of the relaxed-structure eDOS changes when starting from the unrelaxed CIF file. The results are very interesting and underscore the strengths of Mat2Spec. For structures that underwent substantial changes during relaxation, the MAE can be quite large; but for many materials, the Mat2Spec predictions from unrelaxed and relaxed structures are comparable, prompting the following addition to the manuscript and SI:

In Figure S7 we show that for many materials, comparable eDOS predictions can be made using unrelaxed structures, indicating that Mat2Spec may be deployed for predicting the eDOS of any hypothetical material provided the atomic coordinates of each generated structure are sufficiently similar to those of a DFT-relaxed structure to capture the materials chemistry. This mode of deployment of Mat2Spec motivates future study of the sensitivity of eDOS to atomic coordinates and lattice parameters for both DFT-calculated and Mat2Spec-predicted eDOS.

Figure S7. Predicting eDOS of unrelaxed structures: The Materials Project makes available the CIF file of each mp-id prior to the DFT-based relaxation of cell parameters and atomic coordinates. While there are multiple sources of the original “unrelaxed” structures, this provides a non-arbitrary example of an unrelaxed structure for each mp-

id. The eDOS of the unrelaxed structure is not available (never calculated because the typical workflow is DFT relaxation followed by DFT for eDOS). However, the unrelaxed structures can provide some insight in the ability to predict eDOS without DFT-based cell relaxation. Using the test set, the unrelaxed CIF was provided to the trained Mat2Spec SumNorm-KL model to predict the eDOS, with loss calculated using the DFT ground truth of the respective relaxed structure. The results are shown in the top figures as a function of the extent of structure relaxation, (left) the root mean squared (RMS) difference in atomic coordinates and (right) the absolute change unit cell volume. The few points below the break in each horizontal axis correspond to no atomic coordinates or volume relaxation, respectively. As expected, unrelaxed structures that are more different than their relaxed counterparts have a higher propensity for a large MAE. The Pearson correlation coefficients are 0.19 and 0.35 for the RMS distance and volume difference, respectively. To assess whether the unrelaxed structures could be used as a proxy for the relaxed structure for predicting eDOS, the ratio of MAE for Mat2Spec's prediction using the unrelaxed vs. the relaxed structure was calculated for each test set material, providing the distribution shown at bottom. The mode and median are all near a ratio of 1, and the mean is 1.04, demonstrating that on average using the unrelaxed structures increases the MAE by only 4%. As expected, the distribution does not include any very small values for the MAE ratio, while the distribution extends to high ratios, demonstrating that for some materials using the relaxed structure substantially improves the Mat2Spec prediction. The implications for deploying Mat2Spec on unrelaxed structures is that the propensity of high-MAE outliers may increase compared to starting with relaxed structures, although further analysis of the sensitivity of Mat2Spec prediction with respect to atomic coordinates must be performed, especially for the specific subclass of materials where prediction from unrelaxed structures may be deployed. Overall the results indicate that the utility of Mat2Spec will be amplified by development of generative models with reasonable prediction accuracy for the DFT-relaxed atomic coordinates.

- to be extremely clear and honest: outperforming existing (bad) models is not enough. Physics is not about being having a model that is better more often than existing ones, it is rather about getting a model that makes reliable predictions over a well defined domain of applicability. I.e, it is acceptable to have a model that does not cover some cases, but one should know by means of some descriptors which cases are not well (or not at all) described. Plugging everything into a machine-learned model, and the observing a distribution of errors, some of which too large to be of any predictive use is highly not satisfactory. Regardless of how elaborate the learning algorithm and resulting predictive model is.

Response: This is an excellent point with broad implications for machine learning in the physical sciences. We designed the present work as a balance between the desire to solve a prediction task of importance for a specific materials domain and the desire to demonstrate the generality of coupling probabilistic embeddings, contrastive learning, and distribution-based model training for encoding materials chemistry. The diagnostics-focused additions to the manuscript (based

largely on your above comments) will serve the community well in determining the utility of Mat2Spec for a specific domain. Our use case based on gapped metals was designed to demonstrate a specific “domain of applicability” where the domain is specified by a complex set of features in the eDOS. This use case matches the demonstrated scope of eDOS prediction for all periodic structures, but complementary specifications for a sub-domain can be made based on the composition and structure of the materials. As noted above, the distribution of prediction loss can vary substantially among different subclasses of materials, motivating our new study of whether Mat2Spec predictions can be refined for a specific subclass. We made this evaluation for (i) the 3-element materials containing O but not rare earths as well as (ii) materials with spacegroup 225. These subclasses of materials contain a similar number of training examples (2998 and 2325, respectively) but substantially different distributions of loss from the Mat2Spec predictions. We show that by continuing training on each subclass, the respective loss can be improved, which provides an important example mode for deploying Mat2Spec. Furthermore, we show that pre-training on the broader set of materials is a critical step, which further highlights the value of our “global” Mat2Spec model even when researchers are concerned with a small subclass of materials. The new text in the manuscript and new figures in the SI are as follows:

The excellent performance with relatively small training size can also be transformative for materials discovery campaigns conducted within a specific subclass of materials. Using 2 example subclasses from Figure S5, 3-element oxides without rare earths and materials with spacegroup 225, we show that Mat2Spec can be fine-tuned to improve the predictions within each subclass (Figure S8), which enables Mat2Spec to more accurately model specific eDOS features, as shown in Figure S9. Even when one's research is confined to a specific subclass, training Mat2Spec on a broader set of materials helps Mat2Spec encode materials chemistry, as demonstrated by a substantial degradation in performance when using only the data from a specific subclass (Figure S8).

Figure S8 Materials subclass eDOS prediction: The two example sub-classes of materials are (left) materials containing 3 elements including oxygen and 2 non-rare-earth elements and (right) materials with spacegroup 225 (see Figure S5). These classes were chosen to have 1 subclass defined by composition and the other by structure, while producing subclass datasets of comparable size. The size of the train, validation, and test sets is noted, where each set is taken to be the intersection of the sets from the primary eDOS results and the respective materials subclass. For each

subclass, the bars show both the MAE and WD loss for the portion of the original test set that is in the respective sub-class. The results include GATGNN and Mat2Spec in the SumNorm-KL setting with up to 3 different training strategies. The "global" version is trained using the full test set from the primary results in the paper, i.e. not specialized for any specific sub-class, with the loss calculated on the portion of the test set that is within the respective subclass. The "subclass-tune" models are initialized by the respective "global" model with continued training and validation using only the subclass data, which is a type of transfer learning where a model is pre-trained globally and transferred for further training in the sub-domain. For each loss metric and each subclass, Mat2Spec "global" outperforms GATGNN "subclass-tune", and further improvements are obtained by tuning Mat2Spec for the respective subclass. The improvement from the "subclass-tune" training raises the question of whether comparable performance could be obtained without the transfer learning, which was evaluated by the scenario below the dashed line wherein only the train and validation from the subclass was used to train Mat2Spec from a random initialization. The performance is substantially degraded by removing the transfer learning, although even with only (left) 2998 and (right) 2325 training materials, the "subclass-only" training from Mat2Spec still outperforms all versions of GATGNN, providing 2 specific subclass demonstrations of the observation from Figure 4 that Mat2Spec performs well with decreasing training size. The observation that "subclass-only" loss is larger than "global" loss in each case highlights that even when a researcher is focused on a specific materials subclass, training with a broader diversity of materials is better than training locally. Combining these complementary strategies with "subclass-tune" offers additional improvements. Note that for both "subclass-tune" and "subclass-only", each model was also trained in the Standard-MAE setting to provide the normalization factor, i.e. the same methodology described in the main text.

Figure S9 Example eDOS subclass refinements: Using the first subclass from Figure S8, 3-element materials containing oxygen and 2 non-rare-earth elements, example predictions by Mat2Spec SumNorm-KL are shown for a select example in each MAE quintile. Following the protocol for analogous figures, each mp-id is in the same quintile for the 2 model variants, the "global" model from the main text and the "subclass-tune" model that was tuned on this specific subclass. In Q_1 , Q_2 , and Q_3 the "global" model correctly predicts the overall shape, with the "subclass-tune" model better capturing specific features. This is somewhat true in the Q_4 example, although for both Q_4 and Q_5 , both models appear to predict a smoothed version of the eDOS where the high-level structure is captured but not the many high-frequency features (sharp peaks). The

circled regions indicate eDOS features that are better captured by the "subclass-tune" prediction. These examples illustrate the value of the transfer learning strategy described in Figure S8, although some aspects of eDOS for some materials remain imperfectly predicted.

- a minor comment about the methodology: the activation function (eq. 4) should be specified, not just hinted at: "such as the sigmoid". It has been shown that it matters what activation function is used. E.g., the most commonly used these in the recent years is ReLU, but more recently people started tailoring the activation function depending on the task.

Response: Thank you for noting this oversight. We used the SoftPlus function which is a smooth approximation to the ReLU function, and have indicated so in the Methods.

Reviewer #3 (Remarks to the Author):

This paper describes a machine learning model with an architecture tailored to address an important challenge in computational materials science: the prediction of spectra. The proposed approach, named Mat2Spec, predicts spectra (e.g., phonon, electronic densities of state) from material's structure by exploiting the fact that points along these spectra are highly correlated and using a new approach that employs "contrastive learning" to provide further information when training the weights. The work is highly innovative, and the authors take exceptional care in placing the work in context with the community and illustrating its application to materials research. I recommend it to be published with only minor revisions.

Response: Thank you for this apt summary of our work and your support for its publication.

One way I think the paper can be improved is by providing evidence to illustrate the importance of the contrastive learning. The paper establishes the importance of the choice of normalization and training loss function but does not evaluate the effect of other design choices in the architecture. Considering the complexity the contrastive losses add to the approach and, of a lesser importance, the prominence of "contrastive learning" in the title, I would advocate for running some experiments without the contrastive loss or the "alignment" loss of the embedding Gaussians.

Response: This is an excellent suggestion. We performed ablation studies in which (i) the label probabilistic embedding generator was removed to eliminate model alignment during Mat2Spec training, and (ii) the projector was removed to eliminate the supervised contrastive learning. Therefore, the model alignment loss and supervised contrastive loss were also removed, respectively. We performed ablation studies with the eDOS prediction and SumNorm-KL setting. The resulting MAEs are 4.05 and 4.20 states/eV, which are more than 6.6% and 10.5% than that of the original Mat2Spec with the SumNorm-KL setting, respectively. On the other hand, the resulting WDs are 0.24 and 0.27, which are 14.3% and 28.6% higher, respectively, than that of the original Mat2Spec with the SumNorm-KL setting. We have added these results to the manuscript to strengthen the support for our model architecture.

The other recommendation I have for the paper is to clarify a few points of the architecture.

These include:

- What are the dimensionality of the Gaussians? Are they equal to the number of points in the spectrum output?

Response: The dimensionality of the Gaussians is a hyperparameter and is set to 128 in our implementation. To explicitly illustrate this and other dimensions, we have updated Figure 1 to include such information, as also noted in the response to Reviewer #1.

- Correspondingly, does this mean the output embeddings (e.g., Z_F) have a shape of (batch size, number of output points, number of output points)? If so, I can better see how the “translator” and “predictor” layers exploit the correlation between rows in the innermost dimension of the tensor imposed by the Gaussians. If that is a correct line of reasoning, could you describe this logic to the reader? (If not, could you better describe the link between how covariances in the Gaussian propagate to correlations in the predicted labels?)

Response: The shape of Z_f has a shape of (batch size, 128) because Z_f is sampled from the related Gaussian mixture. Regarding the label correlation, it is not modelled by the covariance matrices. To reduce the computational overhead, all multivariate Gaussians have diagonal covariance matrices (cf. the variational autoencoders). The label correlation is in fact modelled by the learned mixing coefficients. The number of Gaussians is equal to the number of output points and the mixing coefficients capture relationships among the points in the spectrum where related points tend to have similar weights. We have incorporated this description into the main text.

- Are the embeddings generated by sampling from the Gaussians?

Response: The embeddings are sampled from the respective Gaussian mixture model.

- Describing in the text how the model is used for inference (i.e., only the embedding produced from the GNN encoding of the material are needed) would be helpful to reinforce the point to the reader.

Response: Fixed. The model inference only needs the feature encoder, representation translator, and predictor, where the feature encoder takes the input materials and produces probabilistic embeddings, the translator translates the probabilistic embeddings into deterministic representations, and the predictor reconstructs the final spectrum properties.

- Explaining the concepts behind the Wasserstein and KL losses would be helpful to non-expert readers to understand why they are better choices than a conventional MSE loss.

Beyond these main comments, I have a few other points of feedback:

Response: Wasserstein and KL losses take into account the probability distribution nature of the spectrum properties which is not considered in the conventional MSE loss. We have briefly discussed this at the end of the “Results” section.

- Briefly introducing the datasets and their usefulness before the Phonon DOS and Electronic DOS sections would improve the reader’s ability to quickly understand the results.

Response: As mentioned broadly in the introduction, spectral properties such as phDOS and eDOS of materials are crucial in understanding their physical and chemical properties. Also, we broadly discuss that the availability of such spectral properties for large sets of materials has recently enabled the use of advanced ML methods for prediction and filtering of new materials. We report the details related to both phDOS and eDOS datasets in the specific “Data generation” section, and per the suggestion we have expanded the introduction of these datasets in the Introduction.

- The use of derived metrics, such as the heat capacity or classification performance, to quantify model performance really help the reader understand the quality of the models. The use of representative cases from the different error quantiles is also a clever approach to showing the model performance in a less-biased way.

Response: Thank you for recognizing the value of these small but key components of our manuscript. We hope that visualizations such as the cross-model error quintiles can be used more broadly in the community for low-bias reporting of ML results.

REVIEWERS' COMMENTS

Reviewer #2 (Remarks to the Author):

The authors have thoroughly addressed my previous comments. Besides the direct replies, the new analysis greatly and importantly expands the discussion of the performance of the proposed Mat2Spec approach.

However, the replies to 3 important points, i.e., physical importance of DFT eDOS, significance of predicting eDOS that are essentially evaluated by the DFT code when the relaxed geometry is found, and possibility of predicting from unrelaxed structures, suggests that the approach is not convincingly providing a compelling use case that promotes it from a technical notable achievement to a useful methodology.

More in detail, eDOS for unoccupied orbitals are typically not accurate enough within DFT (nor there is a physical reason why they should, especially in the Kohn-Sham formulation). The field of ML applied to materials science is definitely moving in the direction of predicting high-level (e.g., GW, Bethe-Salpeter) properties by using DFT geometries and training on few high-level data. Next, it is true that the actual, fine-grid eDOS is calculated as a separate step after the geometry optimization is performed, but I cannot say that Mat2Spec is able to predict accurate, fine-grid-like eDOS on the basis of the input DFT geometries. I.e., the predicted eDOS do not bypass the calculation of the DFT fine-grid eDOS. Finally, as much as the prediction of eDOS via Mat2Spec for converged geometries by inputting non-converged ones shows some promising behavior, it seems to me far to be of practical use, at this point.

In conclusion, I still think that the manuscript is technically very good but not novel/forward looking enough. So, I join Referee 1 with an explicit suggestion to re-submit to a more specialized journal, such as npj Comput. Mater.

Reviewer #3 (Remarks to the Author):

The changes to the paper are great. I now recommend it for publication.

Responses to reviewer comments:

Reviewer #2 (Remarks to the Author):

The authors have thoroughly addressed my previous comments. Besides the direct replies, the new analysis greatly and importantly expands the discussion of the performance of the proposed Mat2Spec approach.

Response: Thank you for guidance with, and recognition of, the substantial improvements to the manuscript.

However, the replies to 3 important points, i.e., physical importance of DFT eDOS, significance of predicting eDOS that are essentially evaluated by the DFT code when the relaxed geometry is found, and possibility of predicting from unrelaxed structures, suggests that the approach is not convincingly providing a compelling use case that promotes it from a technical notable achievement to a useful methodology.

More in detail, eDOS for unoccupied orbitals are typically not accurate enough within DFT (nor there is a physical reason why they should, especially in the Kohn-Sham formulation). The field of ML applied to materials science is definitely moving in the direction of predicting high-level (e.g., GW, Bethe-Salpeter) properties by using DFT geometries and training on few high-level data. Next, it is true that the actual, fine-grid eDOS is calculated as a separate step after the geometry optimization is performed, but I cannot say that Mat2Spec is able to predict accurate, fine-grid-like eDOS on the basis of the input DFT geometries. I.e., the predicted eDOS do not bypass the calculation of the DFT fine-grid eDOS. Finally, as much as the prediction of eDOS via Mat2Spec for converged geometries by inputting non-converged ones shows some promising behavior, it seems to me far to be of practical use, at this point.

In conclusion, I still think that the manuscript is technically very good but not novel/forward looking enough. So, I join Referee 1 with an explicit suggestion to re-submit to a more specialized journal, such as npj Comput. Mater.

Response: While these concerns are valid, they are primarily related to limitations of DFT itself, as opposed to the Mat2Spec model of the present work. We agree that a different eDOS energy grid may be needed for certain applications and have added the following sentence:

“Mat2Spec can also be extended to different energy ranges and energy resolutions, or even to spectra obtained from formalisms beyond DFT, as required for a given materials discovery effort.”

Reviewer #3 (Remarks to the Author):

The changes to the paper are great. I now recommend it for publication.

Response: Thank you for guidance with, and recognition of, the substantial improvements to the manuscript.